# Parallel triplet formation pathways in a singlet fission material

Nilabja Maity [1,6], Woojae Kim [2,5,6], Naitik A. Panjwani [3], Arup Kundu[4], Kanad Majumder [1], Pranav Kasetty[1], Divji Mishra[1], Robert Bittl [3] ✉, Jayashree Nagesh [1] ✉, Jyotishman Dasgupta [4] ✉, Andrew J. Musser [2] ✉ & Satish Patil [1] ✉

Harvesting long-lived free triplets in high yields by utilizing organic singlet fission materials can be the cornerstone for increasing photovoltaic efficiencies potentially. However, except for polyacenes, which are the most studied systems in the singlet fission field, spin-entangled correlated triplet pairs and free triplets born through singlet fission are relatively poorly characterized. By utilizing transient absorption and photoluminescence spectroscopy in supramolecular aggregate thin films consisting of Hamilton-receptor-substituted diketopyrrolopyrrole derivatives, we show that photoexcitation gives rise to the formation of spin-0 correlated triplet pair $^1$(TT) from the lower Frenkel exciton state. The existence of $^1$(TT) is proved through faint Herzberg-Teller emission that is enabled by vibronic coupling and correlated with an artifact-free triplet-state photoinduced absorption in the near-infrared. Surprisingly, transient electron paramagnetic resonance reveals that long-lived triplets are produced through classical intersystem crossing instead of $^1$(TT) dissociation, with the two pathways in competition. Moreover, comparison of the triplet-formation dynamics in J-like and H-like thin films with the same energetics reveals that spin-orbit coupling mediated intersystem crossing persists in both. However, $^1$(TT) only forms in the J-like film, pinpointing the huge impact of intermolecular coupling geometry on singlet fission dynamics.

Singlet fission (SF) is a unique multiexciton generation process in organic semiconductors through which two triplet excitons may be formed from one singlet exciton as long as energy can be conserved[1,2]. The first observation of this phenomenon and its confirmation were made on crystalline anthracene and tetracene, respectively, in the 1960s[3,4]. Meanwhile, there has been an explosion of interest due to the realization that this exciton multiplication process could be used to improve solar cell efficiencies toward 45% (for single-junction Silicon

cells) beyond the Shockley-Queisser limit of 33% by harnessing high energy photons[5–7].

A substantial body of work in the past decade has firmly established a central role in the SF mechanistic pathway for the correlated triplet pair states $^m$(TT)[8–11]. These are multiexciton states consisting of a pair of two triplet excitons which are electronically coupled and spin-entangled. In particular, the $^1$(TT) state, sharing the same multiplicity as the photoexcited singlet, provides the initial gateway to access the

[1]Solid State and Structural Chemistry Unit, Indian Institute of Science, Bangalore 560012, India. [2]Department of Chemistry and Chemical Biology, Cornell University, Ithaca, NY 14853, USA. [3]Berlin Joint EPR Lab, Fachbereich Physik, Freie Universität Berlin, 14195 Berlin, Germany. [4]Department of Chemical Sciences, Tata Institute of Fundamental Research, Mumbai 400005, India. [5]Present address: Department of Chemistry, Yonsei University, Seoul 03722, Republic of Korea. [6]These authors contributed equally: Nilabja Maity, Woojae Kim. ✉e-mail: robert.bittl@fu-berlin.de; jayashreen@iisc.ac.in; dasgupta@tifr.res.in; ajm557@cornell.edu; spatil@iisc.ac.in

triplet manifold on ultrafast timescales through vibronic and/or charge-transfer mediated mechanisms, without requiring any spin flip[2,12,13]. Initially invoked as a theoretical construct to explain triplet-singlet interconversions, $^1$(TT) is now recognized as a distinct electronic state with unique properties such as long-range diffusion, surprisingly rapid non-radiative decay, and symmetry-forbidden fluorescence through a Herzberg-Teller mechanism[14–16]. Following electronic and spin evolution, the higher-spin $^3$(TT) and $^5$(TT) can be accessed[17–19]. Thanks to their differences in spin-allowed decay pathways, the balance between these states is proposed to play a crucial role in determining overall triplet yields[17]. In particular, $^5$(TT) is invoked as the essential final intermediate prior to free triplet formation[20]. These states are all electronically similar, aside from the unique property of $^1$(TT) emission, but their roles can be readily distinguished through a combination of optical (for low-spin TT) and spin-resonance (for high-spin TT) spectroscopic techniques. In particular, the spin polarization patterns of triplets born through SF carry signatures sharply distinct from those of spin-orbit coupling mediated intersystem crossing (SO-ISC) born triplets[21–24], thereby facilitating a clear distinction between SF and SO-ISC born triplets.

This mechanistic picture was chiefly developed on the basis of experiments on tetracene and pentacene derivatives, yet these materials pose a problem for application in real photovoltaic devices due to poor extinction coefficients, low photostability, or energy mismatch to the target bulk semiconductors[7]. Ever more alternative organic materials are theoretically proposed, synthesized, and experimentally characterized to overcome some of the acenes' issues, though often with lower SF rates or yields. One such example recently suggested as a promising SF candidate is the family of diketopyrrolopyrroles (DPP), based on a π-conjugated bicyclic dilactam backbone. The DPP core generally satisfies the energetic prerequisite for SF but also shows high thermal and photostability and a high extinction coefficient[25–27]. Furthermore, the ease of synthetic modification through N-alkylation and aryl substitution permits a high degree of control over solubility, thin-film molecular packing, and electronic energy levels[28,29]. SF dynamics of DPP derivatives have been recently investigated by several groups using transient optical spectroscopy, chiefly transient absorption (TA)[27,30–36]. These studies all report moderately fast and efficient SF, based on the rapid appearance of triplet photoinduced absorption features. Crucially, in DPPs, like several other SF materials including the rylene diimides[37–41], the principal triplet photoinduced absorption band largely overlaps with the ground-state absorption, resulting in derivative-like transient spectra. It is rarely considered that very similar spectral signatures can be produced by pump-induced heating effects causing local refractive index changes–an inevitable effect in typical organic thin films[38,42,43]–severely complicating quantitative analysis. In light of these effects, the close similarities between assigned $^1$(TT) and free triplet optical spectra, and the absence of spin-resonance characterization, there remain significant questions about the spin evolution and triplet-pair dissociation processes in DPPs compared to the acenes.

Here, we newly report HR-TDPP-TEG (Fig. 1a), in which Hamilton receptors (HRs)[44] are linked with 3,6-bis(thiophen-2-yl) DPP (TDPP) through acetylene bridges. Two triethylene glycol (TEG) chains, substituted at the of lactam ring nitrogens and HRs impact the intermolecular packing. Indeed, the HRs permit the formation of self-complementary hydrogen-bonding (via -NH···O=C-) supramolecular aggregate structures. In this work, we obtain two markedly different aggregate coupling motifs (J-like and H-like) in thin film, simply through the choice of solvent used in deposition. Interestingly, these aggregates exhibit almost the same exciton coupling strengths. In contrast to other studies of structure-dependent SF photophysics, this permits an investigation of the impact on SF of molecular packing without any contribution from changes in energetics. Focusing on a characteristic $^1$(TT) photoinduced absorption band in the near-

infrared, which is uncontaminated by thermal effects, we find that the J-like film is capable of SF. The resulting $^1$(TT) state is capable of symmetry-forbidden luminescence–a first among DPP materials. Intriguingly, complementary transient electron paramagnetic resonance (trEPR) measurements demonstrate that, though there is a population of long-lived triplet species, they do not arise from the $^1$(TT) state, which instead quantitatively decay to the ground state. Our optical and spin-resonance data together demonstrate that the DPP film must exhibit parallel triplet formation pathways, i.e., both SF and SO-ISC. This behavior is only followed in the J-like thin film. The H-like film only shows the formation of a low-energy excimer-like species and very slow yields of triplets formed through SO-ISC, highlighting the importance of packing controls in SF materials at a molecular level.

## Results

### Synthesis

As shown in Fig. 1a, HR-TDPP-TEG was synthesized via Sonogashira carbon-carbon coupling reaction. The reaction involves acetylene functionalized HR and respective di-bromo derivatives of TDPP-TEG (TDPP-TEG-Br) in the presence of palladium catalyst Pd(PPh₃)₄ in tetrahydrofuran (THF) and diisopropylamine (DIPA) mixture. The HR was synthesized according to the reported procedure with necessary modifications[45]. A detailed synthetic procedure and structural characterization are shown in Supplementary Note 1 and Figs. 1–4.

### Steady-state optical characterization

In dilute THF, the absorption spectrum exhibits a vibronic progression with a spacing of 0.18 eV (Supplementary Fig. 5), which signifies that the $S_0$–$S_1$ transition of HR-TDPP-TEG is coupled with a C=C stretching mode. The observed 0-0 peak position at 2.02 eV is substantially redshifted from TDPP-TEG (2.27 eV), thanks to the extended conjugation afforded by the HRs[33,46]. This behavior can be further seen in the HOMO and LUMO electronic distributions calculated by DFT (Fig. 1b), which dominate the $S_1$ character. The photoluminescence (PL) spectrum is distinctly narrower with different vibronic structure, indicating that the monomer $S_1$ state is also coupled with a low-frequency torsional mode (typically less than $\omega = 15$ meV) and that structural relaxation along that vibrational coordinate reduces conformational disorder in $S_1$[47]. This overall behavior is consistent with other monomeric DPPs, indicating that our functionalization does not strongly change the photophysics of isolated molecules.

We prepared thin films of HR-TDDP-TEG via drop-casting, obtaining turquoise-colored film **1** from chlorinated solvents (4:1 mixture chloroform (CF): chlorobenzene (CB) or pure CF) and dark-blue-colored film **2** from THF (Fig. 1d). AFM measurements reveal that these film types correspond to completely different microscopic morphologies (Fig. 1d). The film prepared from chlorinated solvents was composed of elongated fibrillar structures with a mean diameter of 50 nm (Fig. 1d, left). In sharp contrast, highly dense ribbon-like microstructures are observed in the film made by THF (Fig. 1c, right).

These different structural motifs are reflected in different intermolecular interactions, as determined from the optical spectra. Compared to the monomer, the steady-state absorption of film **1** exhibits a strongly enhanced ratio of the 0-0 peak to the 0-1 peak intensities: $R_{abs} = A_1/A_2 \sim 1.68$ in film **1**, versus 1.11 in the monomer. Film **2**, on the other hand, reveals the opposite behavior, namely relative suppression of the 0-0 peak, with $R_{abs} \sim 0.69$ (Supplementary Fig. 7). Such peak ratio changes are characteristic of different modes of exciton coupling, and we tentatively assign film **1** as a J-like aggregate and film **2** as an H-like aggregate. To estimate the inter-site coupling strength in thin films, we have analyzed the $R_{abs}$ values in more detail based on the method of Spano and co-workers (Supplementary Note 2)[48,49]. $R_{abs}$ is directly related to the exciton bandwidth ($W$) which in turn yields the exciton coupling strength $J_0$ as $W = 4|J_0|$, as long as we restrict to nearest-neighbor coupling and

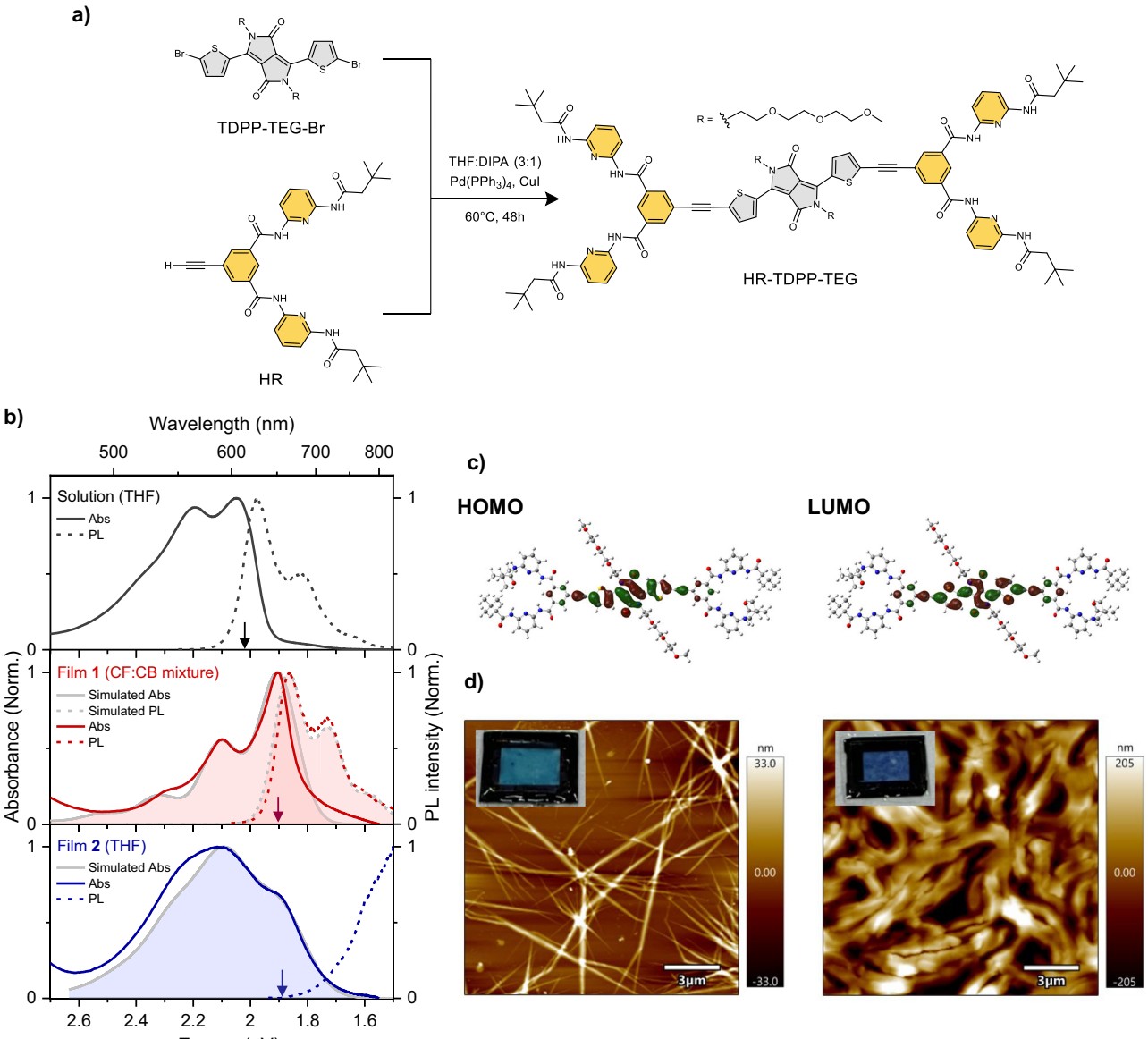

**Fig. 1 | Synthesis and optical characterization. a** Synthesis of HR-TDPP-TEG.
**b** Steady-state absorption (solid lines) and PL (dashed lines) spectra of solution (in THF, top) and two drop-cast films (middle: film **1** prepared with chlorinated solvents, bottom: film **2** prepared with THF) of HR-TDPP-TEG at room temperature. Colored arrows in each panel indicate the position of the 0-0 absorption peak estimated by second derivative analysis (See Supplementary Fig. 5). PL spectra were measured with 2.34 eV (200 fs) photoexcitation and recorded by using an ICCD detector with a 1 μs gate window. Shaded spectra with gray lines indicate simulated absorption (Films **1** and **2**) and PL (Film **1**) spectra based on the site-based one-dimensional Frenkel-Holstein Hamiltonian. **c** Frontier molecular orbitals mainly contributing to the $S_0 \rightarrow S_1$ transition of the HR-TDPP-TEG monomer (isovalue = 0.02 e/(A³)). **d** AFM height images (15 × 15 μm²) for thin films (left: J-like aggregate film, right: H-like aggregate film) of HR-TDPP-TEG. Photos of encapsulated films for optical experiments are also shown as insets, highlighting different colors of the films.

assume that the nature of coupling is purely Coulombic. With the Huang-Rhys factor ($\lambda^2$) and vibrational frequency ($\omega_{vib}$) from the monomer spectra (Supplementary Fig. 6), $W$ values are estimated to be 80 meV for both film **1** and film **2**. These values are found to be smaller than the nuclear reorganization energy due to this vibration ($\lambda^2\omega_{vib}$) of 166 meV, indicating that the HR-TDPP-TEG molecules in thin films are only weakly excitonically coupled. Nonetheless, the spectral shifts for the two films in relation to the monomer are large (120–130 meV, Fig. 1b and Supplementary Fig. 5) and toward the lower energies, in contrast to the expected blue-shift for an H-like aggregate. This effect is due to the gas-to-crystal shift, which here outweighs exciton coupling effects[49–51]. We verify this picture with our simulated absorption and PL spectra using a Frenkel-Holstein Hamiltonian based on the parameters described above and in

Supplementary Table 1. This method yields excellent agreement with the experimental spectra, confirming our analysis from $R_{abs}$. The net result is that the opposite modes of coupling—J-type versus H-type—yield coupled excitonic states that are very similar in energy.

The PL spectrum of film **1** reveals clear vibronic emission with a small Stokes shift and resembles a mirror image of absorption (Fig. 1b, middle). This is in line with typical spectral characteristics of J-aggregates[48]. Nevertheless, given that the equivalent vibronic peak ratio $R_{PL}$ ($I^{0-0}/I^{0-1}$) is smaller in film **1** than in the monomer (Fig. 1b), it can be deduced that the lowest exciton has a localized nature. This result is unsurprising for a system in the weak exciton coupling regime. In contrast, film **2** exhibits energetically broad PL with a large Stokes shift (the peak maximum cannot be precisely estimated due to the detection limit of our ICCD detector). Such a drastic shift implies that film **2**

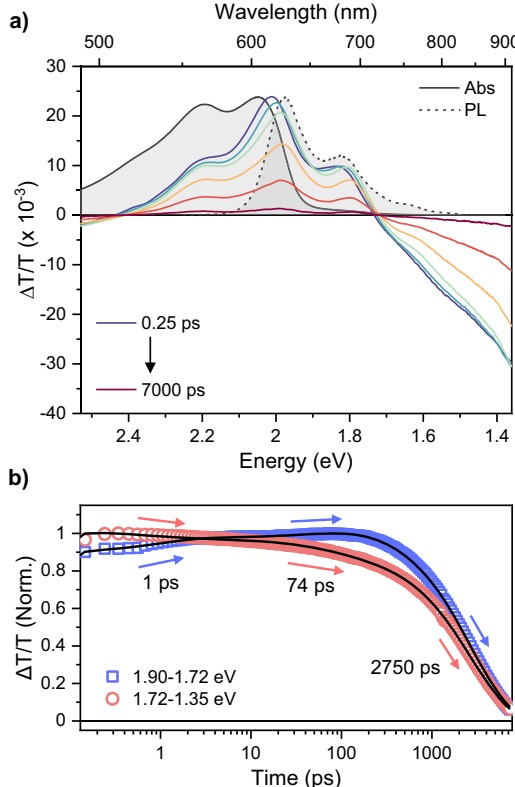

**Fig. 2 | Excited-state dynamics of HR-TDPP-TEG monomer. a** TA spectra of HR-TDPP-TEG in THF after photoexcitation at 2.3 eV with 35.4 μJ/cm² at room temperature. Steady-state absorption (gray solid line) and PL (gray dashed line) are presented for reference. **b** averaged TA kinetics (open symbols) in the two spectral regions (blue−SE region, red−PIA region). The black solid lines are the best global fit of the two kinetics with a triexponential function.

accesses a more relaxed state from the lower Frenkel state, which can be tentatively assigned to the excimer-like state that is widely observed in H-aggregates of various organic materials[52–58]. The PL quantum yields of both films are very low, <1%, compared to 59% in the monomer, suggesting the presence of a fast non-radiative deactivation channel from the emitting state.

## Excited-state dynamics of HR-TDPP-TEG monomer

To understand these non-radiative decay channels, we turn to femtosecond TA spectroscopy. As a baseline for intramolecular photophysics, we present in Fig. 2 the results for HR-TDPP-TEG in THF. After photoexcitation at 2.3 eV in THF solution, we resolve positive ground-state bleach (GSB) and stimulated emission (SE) signals between 2.4 and 1.7 eV and negative photoinduced absorption (PIA) bands above 2.4 eV and below 1.7 eV (Fig. 2a). As delay time increases, the SE bands dynamically redshift, and this corresponds with a slight change of shape of the PIA. This spectral evolution can be ascribed to structural relaxation towards the $S_1$ potential minimum along the torsional coordinate, which occurs biexponentially with the time constants of 1 and 74 ps (Fig. 2b). The $S_1$-state lifetime is estimated to be 2.75 ns, in good agreement with transient PL experiments using the ICCD (2.83 ns, Supplementary Fig. 8). We could not find any triplet signature from monomeric HR-TDPP-TEG (Supplementary Fig. 9), which is in line with its high PL quantum yield (59%) as well as typically low triplet yield (<1%) of other DPP derivatives in the solution phase[33].

## Triplet-like photoinduced absorption

Figure 3a, b shows the TA spectra for film **1** excited at 2.3 eV with a pump fluence of 1.75 μJ/cm², a level at which singlet-singlet

annihilation effects were found to be negligible (Supplementary Fig. 10). Figure 3c highlights the long-time spectral evolution (>1 ns) following excitation at a higher fluence of 53.1 μJ/cm² to improve the signal to noise. The same features are present in the low-fluence experiment but cannot be so clearly discerned (Supplementary Fig. 11).

Immediately after photoexcitation, the transient response in film **1** is dominated by sharp GSB and SE features in the 2.2–1.6 eV region that match the steady-state absorption and PL spectra. These are accompanied by PIA bands towards the edges of our spectral range (Fig. 3a, b). These initial signatures can be readily assigned to the lower Frenkel state due to the presence of well-structured SE. On long timescales (Fig. 3c), the weak residual signal carries a series of derivative-like features in the ground-state absorption region as well as a clear PIA in the 1.7–1.3 eV range. This latter band, though rarely characterized, is a unique marker of triplets in TDPP derivatives[34,59,60], and its presence demonstrates sub-nanosecond (ns) triplet formation. From the TA data alone, it is not possible to determine conclusively whether this state arises from SO-ISC (one triplet exciton) or SF (correlated triplet pair or two triplet excitons). However, on the basis of temperature-dependent PL measurements (see below) we can assign the TA spectra detected after 1 ns to the spin-0 correlated triplet-pair, i.e., $^1(TT)$ state, which is the immediate product of SF. We additionally confirmed by ns-TA that the identical triplet-like PIA in the near-infrared region persists up to microsecond timescales (Fig. 3c), supporting the validity of the assignments above. The detailed discussion on ns-TA data for film 1 can be found in Supplementary Note 3. We also explicitly highlight that we detect spectral evolution between only two electronic species, identified as either lower Frenkel or triplet-related states ($^1(TT)$ or $T_1$), over the entire experimental time window (Fig. 3 and Supplementary Fig. 12). This strongly implies that the direct population of other electronic species, e.g., charge-separated (radical anion and cation) states, in the excited-state dynamics in film **1** can be neglected.

The sub-ns spectral evolution to form $^1(TT)$ is surprisingly rich, exhibiting complex multiexponential behavior rather than a simple mono-exponential conversion between the two states. The dynamics recorded at the principal PIA band from 1.5 to 1.35 eV can be described with a bare minimum of four-time constants: 7, 43, 300, and 1200 ps (Fig. 3d). Only the final time constant has an immediately evident interpretation, as the decay lifetime of $^1(TT)$. To probe the role of excess pump energy to the early-time dynamics, we repeated the experiment using red-edge excitation (1.84 eV) also at low fluence (3.5 μJ/cm²). We obtain a perfect overlap of the kinetics over this wide range of excitation energies (Fig. 3d), revealing that vibrational relaxation or cooling processes make a negligible contribution to the kinetics.

Our kinetic analysis suggests distinct spectral evolution processes prior to 1 ns, and indeed, in each temporal range we detect a unique set of isosbestic points offset from the baseline (Fig. 3b). Such points are a spectroscopic signature of population transfer between two states having different electronic nature. In the initial evolution (7-ps kinetics, 0.2–10 ps panel in Fig. 3b), the GSB intensity barely changes but the SE is strongly quenched. At the same time, the PIA at 1.5 eV from $^1(TT)$ starts to emerge. We thus interpret this process as $^1(TT)$ formation, but the presence of SE beyond the 7-ps window tells us that not all population from the lower Frenkel state undergoes fast $^1(TT)$ formation. In the second and the third processes (43- and 300-ps kinetics, 10–200 ps and 200–1000 ps panels in Fig. 3b), although we still detect isosbestic points offset from the baseline, their positions are slightly shifted, and they are accompanied by a simultaneous drop in the GSB intensity. This combination of features strongly suggests that population transfer from the lower Frenkel state to $^1(TT)$ still occurs but in parallel with significant population loss to the ground state. This behavior would be characteristic of different sub-ensembles within the film showing weaker electronic coupling, and thus slower $^1(TT)$

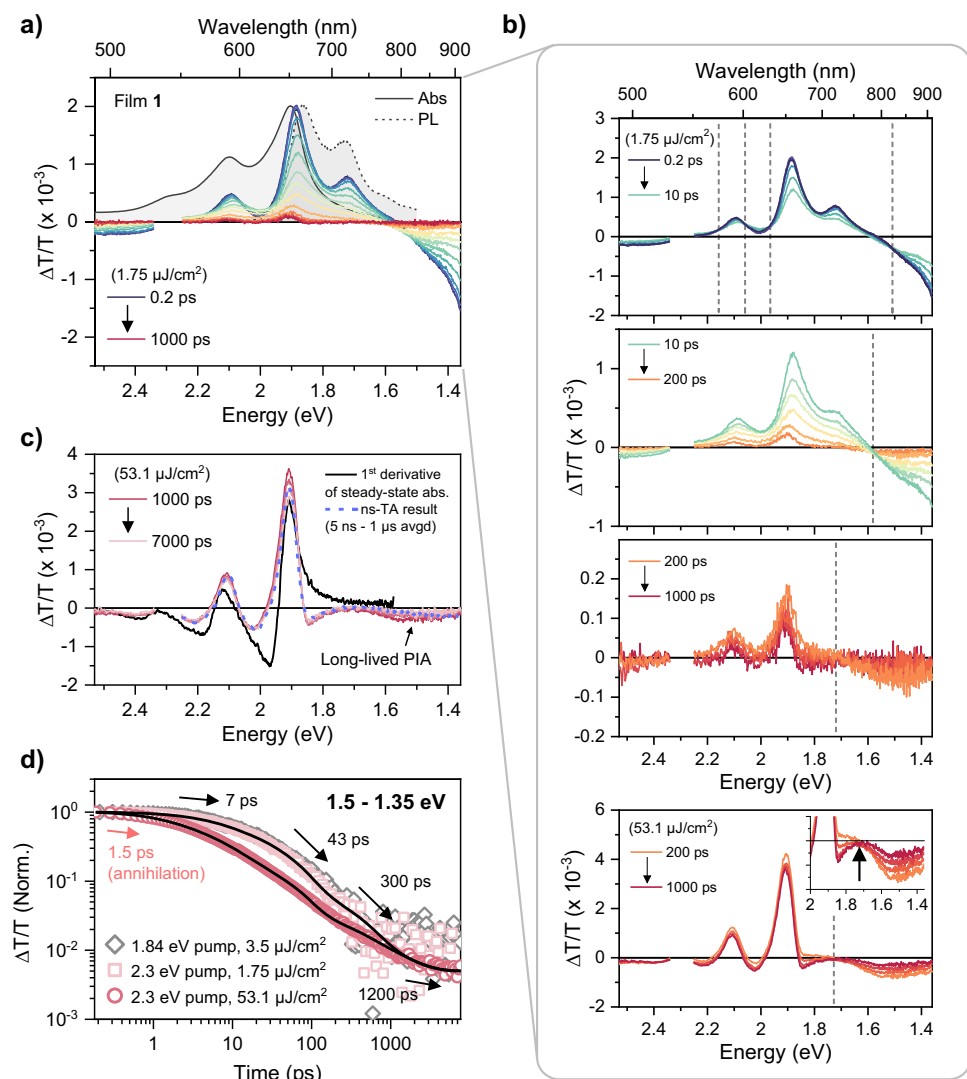

**Fig. 3 | Triplet-like PIA and photoinduced thermal artifacts in film 1. a** TA spectra of film **1** after photoexcitation at 2.3 eV from 0.2 ps to 1 ns with 1.75 μJ/cm². Steady-state absorption (gray solid line) and PL (gray dashed line) are presented in (**a**) for reference. TA spectra near 2.3 eV are truncated due to pump scattering. **b** equivalent to (**a**) but divided into three time regimes to highlight isosbestic points (gray vertical dashed lines) involved in each step of the spectral evolution. The bottom panel is taken from excitation with 53.1 μJ/cm² to show the presence of the same isosbestic point as in the 1.75 μJ/cm² data. **c** TA spectra in the range of 1000–7000 ps with the pump fluence of 53.1 μJ/cm². The first derivative of the steady-state absorption spectrum (black solid line), which is shifted in parallel, is also shown for comparison. **d** Low- (1.75 μJ/cm² for 2.3 eV pump and 3.5 μJ/cm² for 1.84 eV) and high-pump-fluence (53.1 μJ/cm², 2.3 eV pump) TA kinetics averaged in the range of 1.5–1.35 eV. The black lines are the best global fits of the 2.3 eV pump kinetics with a multiexponential function.

formation that is in competition with direct decay from the lower Frenkel state. Overall, the TA data suggest that film **1** undergoes heterogeneous ¹(TT) formation (forming ¹(TT) with different rates) resulting from the microscopic morphological inhomogeneity of thin films.

Transient grating PL (TGPL) results for film **1** provide further insight into the population kinetics of the lower Frenkel state (Fig. 4). To secure a sufficient signal-to-noise ratio of the data, we performed the experiments with a high-pump fluence of about 50 μJ/cm² and confirmed that the integrated PL kinetics are in line with the averaged PIA kinetics in TA obtained with the same pump fluence (Fig. 4b), suggesting that we unambiguously track the kinetics of the same excited-state species, i.e., lower Frenkel. The higher pump fluence data (240 μJ/cm²) additionally support that the sub-ps dynamics stem from singlet-singlet annihilation. After the singlet annihilation, biexponential decays with 8 and 47 ps were obtained through the multiexponential fitting, which is in good agreement with the time constants observed in the TA data. The slower component (300 ps)

was not detectable due to the limit of the delay stage. During the 8- and 47-ps decays, we observe no intensity rise in the kinetics anywhere in our detection region. This allows us to rule out relaxation along the same exciton manifold, which typically presents as a dynamic redshift in time-resolved PL spectra with a decay at higher energy and a rise at lower energy[61]. We note, however, that the peak positions (~10 meV difference) and vibronic peak ratios on these characteristic timescales clearly differ (Fig. 4c), implying a slightly different electronic structure between the two. Considering as well the substantial quenching in emission intensity that these decays reveal, and the close agreement with TA kinetics, we find that initial time constants reflect parallel ¹(TT) formation from a distribution of lower Frenkel states.

We have carried out equivalent TA experiments on H-like film **2** as well. In contrast to film **1**, even at the earliest timescales, we observe no SE bands, consistent with the weak emitting properties of the H-type coupled state (Fig. 5a). The response is instead dominated by strong GSB and PIA bands and bears a resemblance to the behavior of other

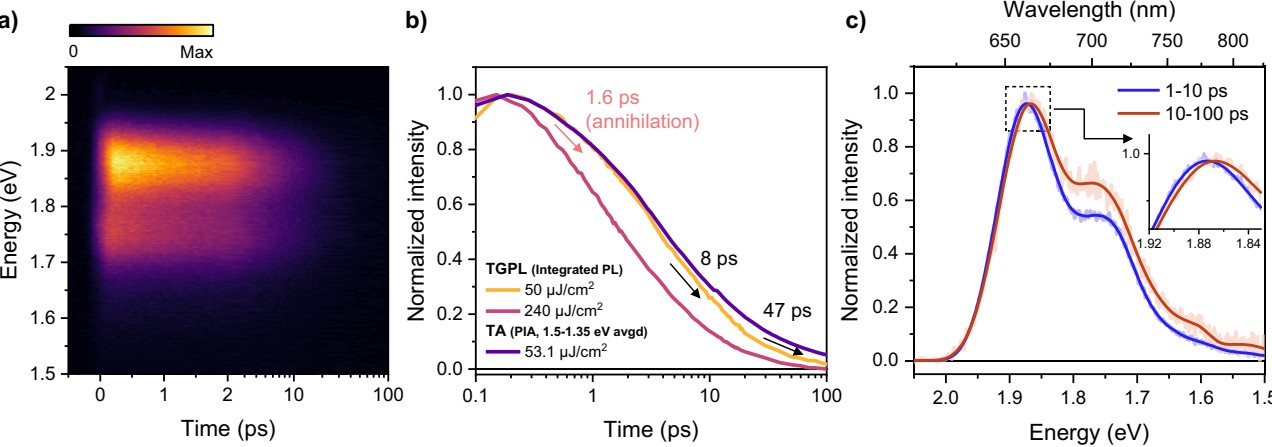

**Fig. 4 | Lower Frenkel PL kinetics of film 1. a** TGPL map of film **1** following photoexcitation at 2.1 eV with a pump fluence of ~50 μJ/cm². **b** Comparison of TGPL (integrated PL) and TA (averaged PIA) kinetics with different pump fluences. The time constants in the panel were obtained by fitting the 50 μJ/cm² integrated PL

kinetics (yellow line) with a multiexponential function. **c** Comparison of normalized PL spectra (light color−raw, dark color−smoothed) averaged over two different time regions (1–10 ps vs. 10–100 ps) in the data measured with 50 μJ/cm². The inset shows the difference in peak positions between the two spectra.

organic molecules forming H-aggregates and showing excimer-like states[55,58]. Also unlike film **1**, we detect no significant spectral evolution or isosbestic points over the full decay range. Although film **2** also reveals long-lived signals into the nanosecond regime, no PIA features are in the NIR region (Fig. 5b), only a surprising positive band (see below). Using a similar analysis of the kinetics averaged from 1.8 to 1.35 eV, we obtain a multiexponential decay with time constants of 12, 84, and 363 ps. Given the lack of spectral evolution, we speculate that the transition from the initial Franck-Condon state to the excimer-like state is faster than the IRF of our setup (200 fs), as other excimer-forming systems, because the monomers are already stacked in the ground state[58]. A detailed spectral analysis of film **2** is presented in Supplementary Fig. 12. Crucially, there is no evidence of any timescale of the triplet-like PIA in film **2**, implying that the ¹(TT) formation channel is blocked. Furthermore, we were not able to obtain TGPL signals from film **2**, which is presumably due to the very weakly emitting nature of the excimer-like state.

## Photoinduced thermal artifacts
We note that both films exhibit derivative-like TA spectra in the region of the ground-state absorption, most prominently on longer time-scales (Figs. 3c and 5b). This effect can arise from overlapping contributions of GSB and PIA. In many reported SF materials, particularly rylene and DPP derivatives, the most intense triplet-state PIA overlaps with the GSB, and consequently, such spectra are typically regarded as a hallmark of triplet exciton formation. While sensitization experiments confirm that such spectral shapes can arise from triplets, we note that remarkably similar features can be generated by pump-induced transient thermal effects, which are insensitive to excitation power or repetition rate and thus behave like electronic population signatures in most crucial respects[38,42,43]. One simple method to check their impact is to compare the long-lived TA spectrum and the first derivative of the absorption spectrum[42]. As can be seen in Figs. 3c and 4b, the line shapes of the two spectra are well-matched with each other for both film **1** and film **2**, suggesting a significant contribution of thermal artifacts to the TA signals in the ground-state absorption region. The presence of these signals even at very low pump fluence, and their persistence throughout the ns-μs regime[38,39,42], significantly complicates any quantitative analysis of our data, for instance to calculate the triplet yield based on the singlet depletion method. To avoid any misinterpretation, we solely analyze the PIA band in the NIR region which is not overlapped with the GSB. Even here, caution is necessary: thermal signals can pull up NIR PIA bands towards positive ΔT/T, due

to samples reflectance changes induced by the thermal change of refractive index[42,62,63]. This case is prominent in film **2** (Fig. 5b), thanks to the absence of any long-lived PIA in the NIR.

## One vs. two emissive states
We gain further insight into the triplet formation process from time-resolved PL spectroscopy on the ns timescale, which is not affected by such thermal artifacts and provides greater sensitivity over a wider dynamic range than TGPL[38]. Figure 5 shows time-resolved PL results for films **1** and **2** obtained after 2.34 eV excitation with low pump fluence (2 μJ/cm²). We used the same 200 fs pump pulses as in TA, but here the time resolution is limited by the electronic gate of our ICCD detector to approximately 1 ns (a flat-top like IRF from −500 to 500 ps, Supplementary Fig. 13). In the case of J-like film **1**, intense vibronic PL akin to the steady-state spectrum appears at time zero. Beyond the IRF, the signal magnitude is reduced by ~99% similar to what we observed in TGPL, with weak delayed emission persisting up to 10 ns (Fig. 6a and Supplementary Fig. 14). Surprisingly, the spectral shape of the weak delayed PL strongly differs from both the prompt PL and the steady-state PL, showing a reversed vibronic peak ratio (Fig. 6c). Analyzing the spectral slices with area normalization, we can observe a clear iso-emissive point near 1.7 eV (Fig. 6e). This interesting behavior suggests that there are two distinct emissive states contributing to the time-resolved PL kinetics[64]. Moreover, we recall that the only electronic signatures detected in TA on these timescales are those for triplet excitons. This behavior indicates that the presence of triplets results in a distinct radiative decay pathway. In contrast to film **1**, film **2** does not show any notable spectral evolution, namely the spectral shape of early and later delay times is nearly the same (Fig. 6b, d). Accordingly, the area-normalized analysis shows no isoemissive points, confirming that there is only one emissive state in film **2** (Fig. 6f).

## ¹(TT) emission
To understand the origin of the two emissive species in film **1**, we repeat the experiment at a range of temperatures. Consistent with their assignment to different electronic states, the prompt and the delayed PL show distinct responses. Upon cooling, the prompt PL reveals an enhanced 0-0 peak relative to 0-1 vibronic emission (Fig. 7a). This behavior is in good agreement with the temperature-dependent optical response of J-aggregates[65], and it can be related to superradiance owing to reduced thermal disorder at low temperature. It provides an unambiguous demonstration that the prompt emission arises from the photoexcited lower Frenkel state. In the delayed PL, decreasing the

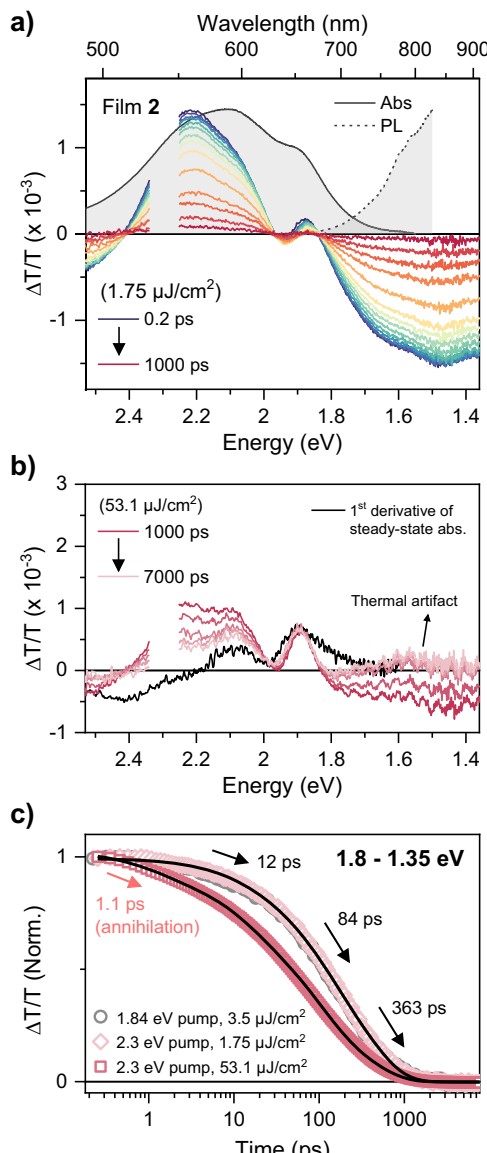

**Fig. 5 | Excimer-like PIA and photoinduced thermal artifacts in film 2. a** TA spectra of film **2** after photoexcitation at 2.3 eV from 0.2 ps to 1 ns with 1.75 μJ/cm². Steady-state absorption (gray solid line) and PL (gray dashed line) are presented in **a** for reference. TA spectra near 2.3 eV are truncated due to pump scattering. **b** TA spectra in the range of 1000–7000 ps with the pump fluence of 53.1 μJ/cm². The first derivative of the steady-state absorption spectrum (black solid line), which is shifted in parallel, is also shown for comparison. **c** Low- (1.75 μJ/cm² for 2.3 eV pump and 3.5 μJ/cm² for 1.84 eV) and high-pump-fluence (53.1 μJ/cm², 2.3 eV pump) TA kinetics averaged in the range of 1.8–1.35 eV. The black lines are the best global fits of the two 2.3 eV pump kinetics with a multiexponential function.

temperature actually suppresses emission at the 0-0 peak and the lower vibronic progressions dominate the spectra (Fig. 7b). Together with the presence of only triplet-like PIA, this behavior is in excellent agreement with the previous reports of temperature-dependent ¹(TT) emission in (hetero)acenes, zethrenes, and conjugated polymers[15,16,66,67]. In principle, emission from the ¹(TT) state to the ground state is symmetry-forbidden. However, the transition can become partially dipole-permitted by borrowing oscillator strength from a nearby state through Herzberg-Teller vibronic coupling, with the aid of non-totally symmetric vibrations[68,69]. It has been known that this mechanism is also activated in carotenoids[70], cyclic conjugated oligomers[47,71], and some polycyclic aromatic hydrocarbons[72], in which their $S_1$ states commonly have one-photon-forbidden nature. In this framework, the strong

temperature dependence at the 0-0 peak is a signature of thermally activated repopulation of emissive lower Frenkel, and it suggests a small activation barrier of 33 meV (Supplementary Fig. 16)[15,16]. Alternatively, following previous methods we can add one vibrational quantum to the 0-1 peak position of the ¹(TT) emission to estimate the ¹(TT) energy as 1.83 eV, compared to a singlet lower Frenkel energy (from PL) of 1.85 eV (Supplementary Fig. 16)[16]. Such weakly exothermic SF, similar to the case of $F_2$-TES-ADT (an anthradithiophene derivative), tetracene, TIPS-tetracene, and rubrene, helps to understand the heterogeneity of SF rates seen in Figs. 3 and 4, and the coexistence of singlet and triplet features over moderate time delays.

Despite the suitability of DPP dyes for SF and repeated study with TA spectroscopy[32,33], ¹(TT) emission has never been reported in this family of materials. This first and direct observation of ¹(TT) emission in a DPP thin film allows us to conclude that the triplet-like PIA observed in the TA data stems from SF, not from SO-ISC. This observation highlights the importance of directly correlating time-resolved PL with TA to understand SF and especially triplet-pair formation. Often the triplet features in TA are not prominent or are obscured by thermal effects, and very frequently the spectral differences between bound (i.e., ¹(TT)) and free triplets are too slight for reliable determination. Symmetry-forbidden emission in these instances provides a unique, often background-free signature of the ¹(TT) state.

Crucially, the delayed ¹(TT) emission decays at room temperature with a time constant of 1.2 ns, which is quintively in line with the TA kinetics in the ¹(TT) PIA band at the NIR region from 1 to 7.6 ns (Fig. 6c and Supplementary Fig. 17). On lowering the temperature, the lifetime of ¹(TT) emission is enhanced, presumably due to suppression of other non-radiative deactivation channels at low temperature, but we still observe essentially complete decay within 50 ns. For comparison, we note that there are no significant temperature-dependent spectral changes in film **2** (Fig. 6b). We only detect slight bandwidth changes, implying that energy distribution on the potential energy surface of the excimer-like state is modulated controlled by temperature. As in film **1**, cooling increases the delayed emission lifetime, which we again ascribe to a reduction in non-radiative deactivation rates.

## Long-lived triplets born through SO-ISC

To learn more about the fate of ¹(TT) we carry out trEPR spectroscopy on film **1** and, for comparison, film **2**. We focus our analysis of film **1** on the spectrum acquired at 50 K with excitation at 1.91 eV (Fig. 8a), though we observe negligible excitation energy or temperature dependence. Our measurements show a weak but clear triplet state spectrum with a polarization pattern of *eea/eaa* (where *e* is emission and *a* is enhanced absorption). This spectrum can be described by zero-field splitting parameters (ZFS) of $D \approx 1150$–1250 MHz and $|E| \approx 290$–360 MHz, with the uncertainty dominated by the low signal level and partial ordering effects. Our $E$ value is in good agreement with previously determined ZFS parameters for the triplet in a related TDPP derivative, formed via charge recombination in a blend film with $PC_{70}BM$. The reported $D = 1550$ MHz is distinctly higher[59], suggesting that the extended side chains in HR-TDPP-TEG permit greater delocalization of the spin density. Crucially, the observed polarization pattern of eea/eaa indicates the mechanism for triplet formation in film **1** is SO-ISC. The inner canonical peaks are not apparent when excitation is at 2.33 eV, suggesting a slightly different zero-field triplet sublevel population. The polarization pattern whether *eea/eaa* or *eee/aaa* corresponds to an SO-ISC-born triplet, that is, the triplets observed on these timescales do not arise from SF. This can be seen unambiguously by comparison to model spectra for the two pathways (Fig. 7b), where SF-born free triplets exhibit a unique polarization pattern due to direct population of the high-field eigenstates (*aee/aae* for $D > 0$)[18,19]. See Supplementary Note 6 for further details on the model spectra.

To extend our search for SF signatures, we repeated these measurements at different temperatures. At 20 K, in addition to increased

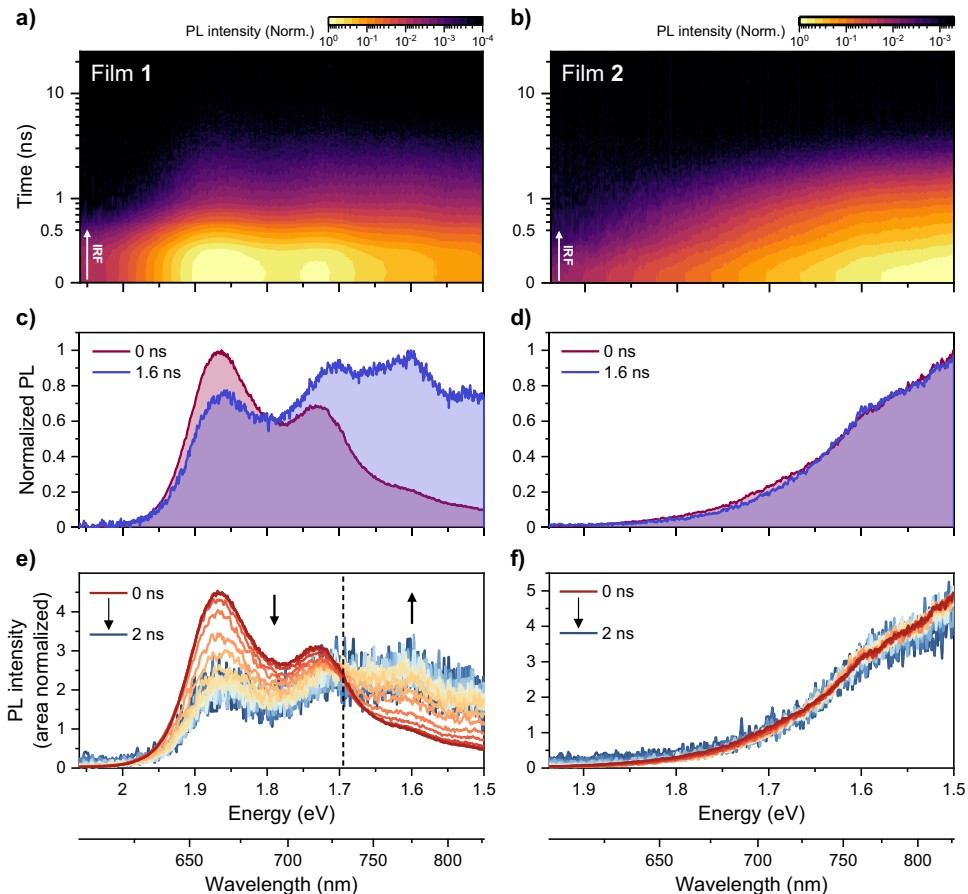

**Fig. 6 | One vs. two emissive states.** Time-resolved PL results for **a, c, e** film **1** and **b, d, f** film **2** after photoexcitation of 2.34 eV at 2 μJ/cm². **a, b** Two-dimensional contour maps of the time-resolved PL data. **c, d** Comparison of intensity-normalized time-resolved PL spectra at the specified delay times. Because of low signal intensities, the spectra at 1.6 ns were measured in a separate experiment to achieve a sufficient signal-to-noise ratio. **e, f** Area-normalized time-resolved PL spectra from 0 to 2 ns with 0.1 ns interval. The dashed line in the left bottom panel for film **1** indicates an isoemissive point near 1.7 eV, supporting the presence of two emissive states.

triplet lifetime and spin-relaxation times there is precedent to observe long-lived $^5$(TT). This state, unique to SF systems, yields additional features with a spectral width of D/3. Yet we observe no significant triplet spectral changes at 20 K (Supplementary Figs. 22 and 23). At 290 K the conditions approximate those used for time-resolved optical spectroscopy, and again we detect no difference in the extremely weak triplet polarization pattern (Supplementary Figs. 24 and 25). As in the optical experiments, we find that excitation energy has no effect. Strikingly, we even observe similar triplet features—polarization pattern and spectral width—in H-like film **2**, which shows no evidence for SF and no detectable optical signatures of triplet formation. Because of this surprising behavior, we additionally considered the role of orientation of both samples relative to the external magnetic field, which has been shown to impact the trEPR spectra of triplets in other SF materials[73]. In neither film did we observe the appearance of SF spectral signatures at other angles. Full details on the trEPR measurements at different temperatures, rotation angles, and pump energies are provided in the Supplementary Information. From this lack of dependence on temperature, excitation energy, orientation, or even intermolecular coupling motif, we can unambiguously conclude that the evolution of $^1$(TT) into free triplets is not a preferred pathway in film **1**. In both films, the very small population of triplets must arise from SO-ISC. Combining with ns-TA results, although it is not possible to precisely quantify the triplet yield due to the thermal artifacts, we highlight that the yield from the SO-ISC pathway is likewise evidently very small (Supplementary Fig. 12d). Furthermore, a CT state signal is present in both samples at low temperatures (See Supplementary Note 7).

## Discussion

Our combined time-resolved electronic and spin-resonance spectroscopic results of the HR-TDPP-TEG films lead to the interesting conclusion that the long-lived triplets observed in an SF material need not stem from SF – SO-ISC can present a viable parallel relaxation pathway. The results also point out a much more complicated situation of SF kinetics, contrary to what has been reported so far in DPP derivatives. This is obviously an undesirable situation for photovoltaic applications, and it highlights the great care needed in characterizing SF in the solid-state. Because of the intrinsically spin-entangled nature of the triplet pair, using either electronic or spin-resonance spectroscopy alone can only reveal one side of the SF dynamics: it is, in most cases, impossible to distinguish the states having different spin quantum numbers with electronic spectroscopy, whereas the spin-0 $^1$(TT) state is dark to spin-resonance methods.

We find that such caution is particularly warranted for systems in which the TA signatures of triplets largely overlap with the ground-state absorption and thus thermally induced effects. In DPP systems, the near-infrared triplet PIA provides a more robust route to study triplet-related species, particularly in concert with PL spectroscopy to capture the unique signatures of $^1$(TT). Still, the weakness of this band makes quantification challenging. It is likely that similar to the case of the acenes[19,74–82], experiments on tailored dimers in solution (where

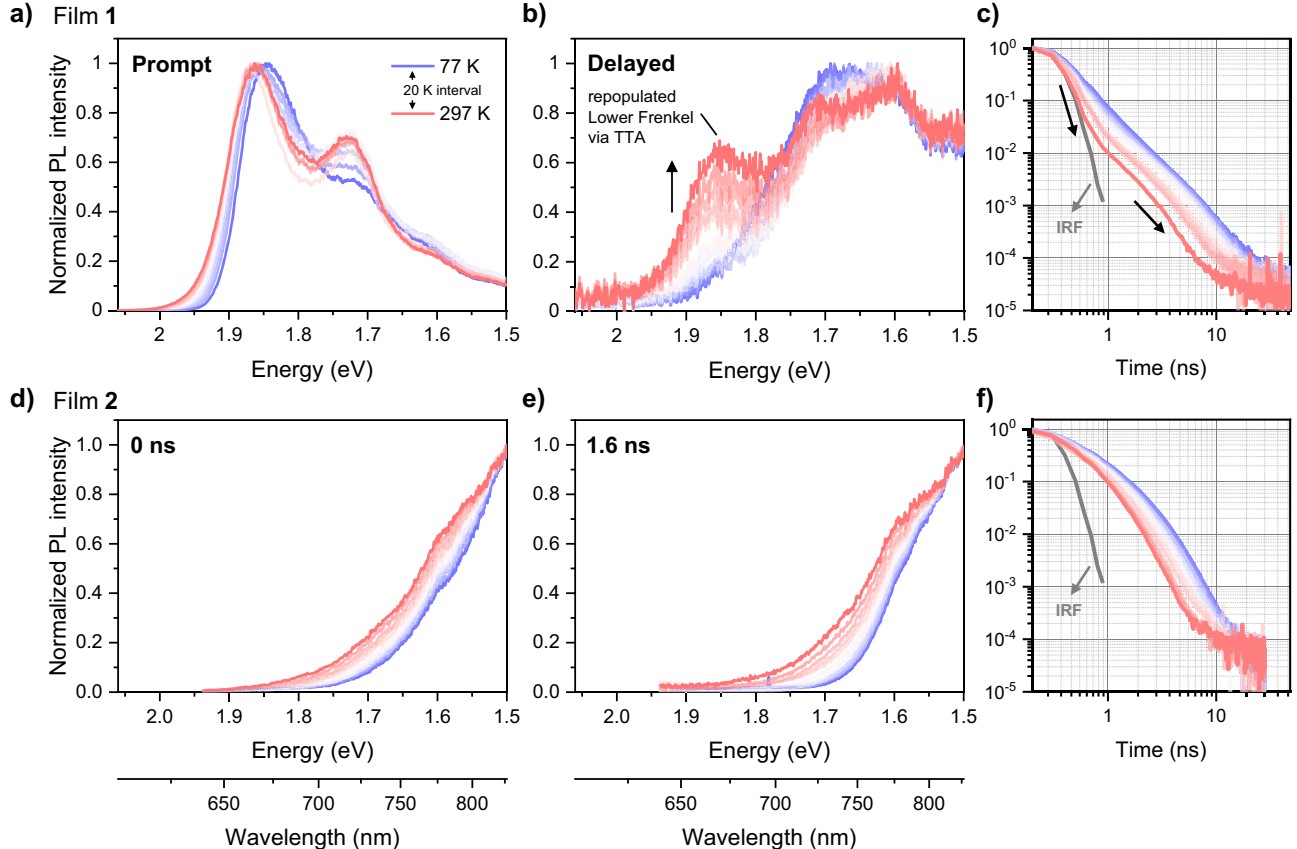

**Fig. 7 | $^1$(TT) emission.** Temperature dependence of time-resolved PL in **a**, **b**, **c** film **1** and **d**, **e**, **f** film **2** after photoexcitation of 2.34 eV at 2 μJ/cm². The data were collected from 77 K to 297 K with 20 K interval. **a**, **d** and **b**, **e** representative normalized time-resolved PL spectra at early and later delay times, respectively. For film **1**, prompt PL spectra are taken from 0 ns, and delayed PL spectra were obtained by averaging all spectra showing peak intensity <10$^{-2}$ in the normalized data. The black arrow in the

middle panel for film **1** highlights the growth of the 0-0 band of the repopulated lower Frenkel emission via triplet-triplet annihilation (TTA) upon rising temperature. Film **2** does not show the distinct prompt/delayed behaviors so early and later spectra were taken from 0 and 1.6 ns, respectively. Panels **c** and **f** indicate integrated PL kinetics.

thermal effects are negligible due to rapid heat dissipation) could provide a detailed mechanistic insight into the SF pathway in DPPs. To date, though, such structures have tended to exhibit only symmetry-breaking charge transfer or SO-ISC, without any $^1$(TT) formation[83,84].

A widespread result in intramolecular SF systems is that $^1$(TT) can be readily formed, but confinement within the dimer limits the evolution into long-lived triplets. Entropic effects in thin films can be invoked to explain the difference between solid-state and dimeric samples, but our spin-resonance studies reveal the limitations of that idea. Despite the evident fast formation of $^1$(TT) in HR-TDPP-TEG thin films, it does not dissociate into free triplets. We note that while SF has been reported in numerous TDPP-based thin films, none have included trEPR confirmation for the SF pathway, a vital measurement. However, the electronic kinetics reported in Fig. 3 are broadly similar to all previous TDPP SF, including in the magnitude of overall signal decay within a few ns. This leads us to speculate that $^1$(TT) mostly exhibits rapid and efficient decay directly to the ground state (principally non-radiative, but also radiative) not only in HR-TDPP-TEG J-type aggregates but across the TDPP family. The reason for such a low yield of dissociation into free triplets is currently unclear. Potentially the $^1$(TT) binding energy is so large that $^5$(TT) is effectively inaccessible, leading to the triplet pair behaving like a singlet exciton with no spin evolution. It should be possible to overcome this effect in thin films through triplet hopping to reduce the inter-triplet interactions[18], but our temperature-independent trEPR suggest even this pathway is inactive. In order to understand this in more detail, future work, for example,

magnetic-field-dependent experiments to manipulate energy levels of $^m$(TT) are further required.

Beyond the nuances of SF spectroscopy, the HR-TDPP-TEG system provides a novel platform to probe the role of intermolecular coupling. In contrast to other systems[38,85–87], the coincidence of exciton coupling strengths in our J-like and H-like films means that the energetic landscape for SF is essentially the same. Thus, our results are uniquely sensitive to the degree and nature of intermolecular orbital overlap. We find that the subtle differences between these two weakly coupled motifs are sufficient to enable or completely disable $^1$(TT) formation. This finding validates the common perspective that the slip-stacked J-aggregate geometry is better suited to SF than the face-to-face H-aggregate geometry, providing markedly enhanced coupling between the lower Frenkel and $^1$(TT)[2]. The lower-lying excimer-like state observed in the H-like film may contribute as well, making SF distinctly endothermic on longer timescales once relaxation is complete. There has been continuous debate on the possible role of such excimers in the SF pathway[88]; since they can be viewed as an adiabatic mixture of locally-excited and charge-transfer wavefunctions, they can potentially provide the enhanced couplings invoked in models of mediated SF. However, where this wavefunction mixing is accompanied by strong π overlap and thus stabilization, the net effect is likely to be a transition to an endothermic process, i.e., a trap state. Comparison of our two DPP packing motifs from this perspective supports the idea that excimer-like species are generally detrimental to $^1$(TT) formation. Some highly exothermic systems may be robust to this

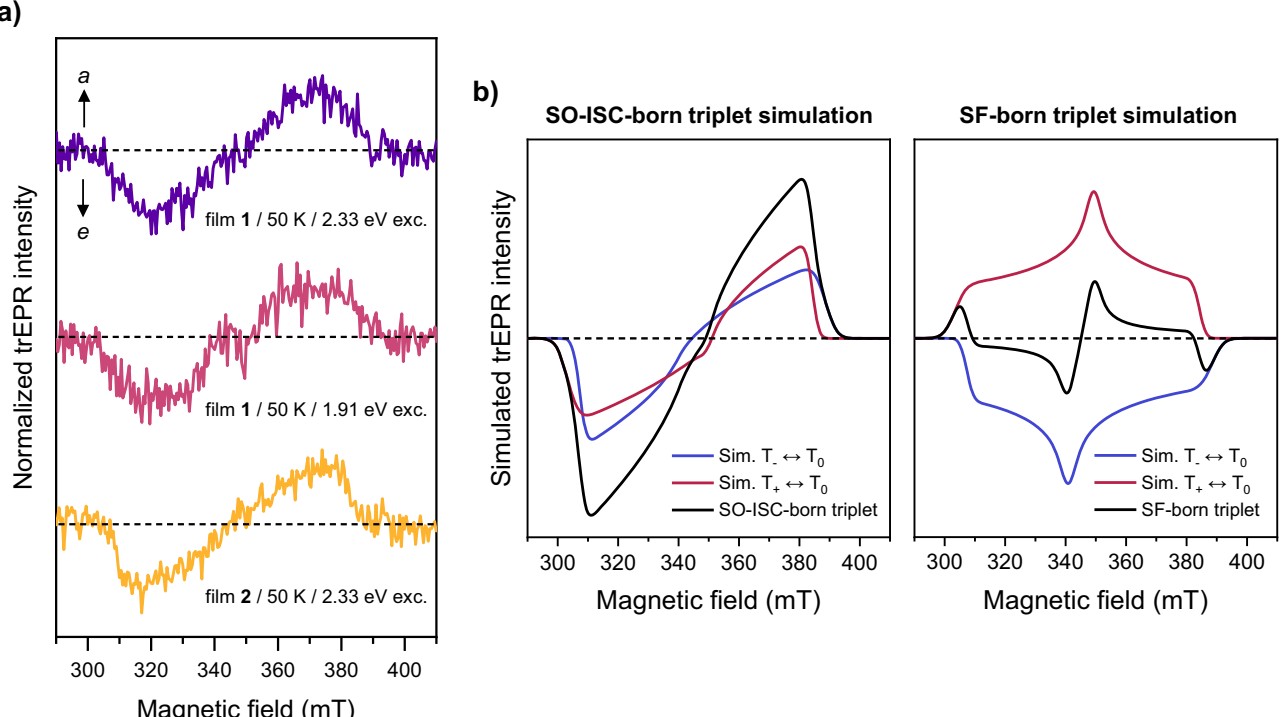

**Fig. 8 | SO-ISC-born long-lived triplets. a** Averaged trEPR spectra (0.1–2.8 μs) of film **1** and film **2**. Temperature and applied excitation energy are indicated below each spectrum. All spectra were measured with an angle of 0° relative to the external magnetic field direction. See Supplementary Information for further details. **b** Representative simulation of SO-ISC- and SF-born triplets using EasySpin with ZFS parameters $D = 1200$ MHz and $E = -320$ MHz. The blue and red lines represent the simulated spectra of the $T_- \leftrightarrow T_0$ and $T_+ \leftrightarrow T_0$ components, clearly showing the polarization pattern.

effect (e.g., hexacene), but in general a more viable approach to harness the charge-transfer coupling in H-aggregates will be molecular engineering to achieve null aggregation[89,90].

## Methods

### Materials
Details of the synthesis and characterizations of HR-TDPP-TEG are reported in the Supplementary Information (Supplementary Note 1).

### Sample preparation
Samples were prepared by drop casting 2.5 mg/ml solutions of HR-TDPP-TEG prepared in either chlorinated solvents ($CHCl_3$ or $CHCl_3$/chlorobenzene mixture) or THF (anhydrous) onto precleaned quartz substrates. Comparison of steady-state absorption spectra of thin films, for optical and EPR spectroscopy, fabricated in different laboratories is shown in Supplementary Fig. 18.

### Atomic force microscopy
Morphology of the self-assembly was probed using atomic force microscopy (AFM). The AFM image for all films was recorded using Oxford instruments Asylum research Cypher instrument in tapping mode with a 10 nm radius Si tip scanning with 40 N/m force. RMS roughness and the height images were processed in the Cypher 16.05 software.

### Theoretical modeling
Details of theoretical modeling for the steady-state spectra of HR-TDPP-TEG thin films are reported in the Supplementary Information (Supplementary Note 2).

### Steady-state and time-resolved optical spectroscopy
The steady-state absorption spectra in solution and thin films were recorded with a home-built absorption spectrometer based on

Avantes miniature spectrometer (Avaspec-Mini-4096CL). The steady-state and time-resolved PL spectra were measured with an ICCD detector (PI-MAX4, Princeton Instruments) after photoexcitation using 2.34 eV pump (200 fs) generated through an optical parametric amplifier (OPA) and its second harmonic (SH) module (ORPHEUS and LYRA-SH, Light Conversion), pumped by a Yb:KGW amplifier (PHAROS-SP, Light Conversion). For steady-state PL, the repetitive mode with a 1 μs gate window was used, whereas, for time-resolved PL, the sequential mode with a 0.48 ns gate window was utilized. For temperature-dependent experiments, the samples were cooled in a liquid nitrogen cryostat (Janis VPF-100, Janis Research Company LLC) under vacuum conditions (~$10^{-6}$ mbar). PL quantum yields of the samples were measured by absolute method, using an integrated sphere of Quanta-Phai instrument. Transient absorption measurements were performed with an automated transient absorption spectrometer (HELIOS, Ultrafast Systems), also driven by the Yb:KGW amplifier (PHAROS-SP, Light Conversion) operating at 8 kHz. The same OPA and SH module (ORPHEUS and LYRA-SH, Light Conversion) used in the PL experiments generate a 200-fs narrowband pump pulse. A portion of the fundamental was separated to generate a white light continuum probe pulse ranging from 2.53 to 1.35 eV using a 1 cm YAG crystal. The beam diameters (1/$e^2$ height) for pump and probe pulses at the sample position were 650 and 200 μm, respectively. TA spectra were collected with magic angle condition between pump and probe and in a shot-to-shot fashion. Pump-probe time delay was set by a mechanical delay stage from −3 to 7600 ps. For studying sub-100 ps PL, we utilized a home-built transient grating PL spectrometer of which technical detail is described in Chen et al.[91]. In brief, the same OPA and SH module driven by the Yb amplifier (8 kHz) used in the TA spectrometer were utilized for the pump, and its beam diameter (1/$e^2$ height) at the sample position was approximately 55 μm. The PL from the sample was collected by a reflection geometry with an off-axis parabolic mirror, which enabled us to ignore the inner filter effect. For

magic angle experiments, the pump polarization was set to a magic angle to horizontal polarization by a half-wave plate (AHWP05M-600, Thorlabs), the collected PL was horizontally filtered by a wire-grid polarizer (WP25M-UB, Thorlabs) and focused by an off-axis parabolic mirror. For gate pulses, two identical horizontally polarized fundamental (1030 nm) beams of ~20 µJ were made by a 50:50 (R:T) beam splitter and were focused on a 2 mm YAG plate by an achromatic lens with an external angle of ~4.2° to generate transient grating allowing us to collect PL ranging from 400 to 850 nm simultaneously without changing optics geometry. Pump-gate delays were set by a linear translational stage (LTS300, Thorlabs) and diffracted PL was collected by an ICCD detector (PI-MAX4, Princeton Instruments). The data acquisition was done with a home-built LabVIEW code. The nanosecond to microsecond transient absorption data were collected using a commercial pump-probe system (EOS, Ultrafast Systems), having ~800 ps time resolution. For the excitation pulse, the 1 kHz repetition rate femtosecond pulses obtained from the OPA was used. The white light supercontinuum probe (360–1600 nm, with spectral resolution of 1.5 nm (VIS) and 3.5 nm (NIR)) was generated by focusing a Nd:YAG laser pulse into a photonic crystal fiber. The probe pulses were electronically synchronized with the femtosecond regenerative amplifier, and the pump-probe delay time was controlled by a digital delay generator (CNT-90, Pendulum Instruments). To minimize the noise, the probe beam was referenced with respect to the signal channel. All thin films for time-resolved spectroscopic experiments were prepared and encapsulated under the nitrogen atmosphere (Fig. 1d). No sample degradation was observed throughout the experiments.

### Transient electron paramagnetic spectroscopy

Transient EPR experiments were performed on a laboratory-built X-band (9.7 GHz) continuous wave spectrometer together with a Bruker MD5 dielectric ring resonator with optical access. Optical excitation at 2.33 eV (532 nm) was provided using a diode-pumped Nd:YAG laser (Atum Laser Titan AC compact 15 MM) equipped with a second harmonic generator, with an incident pulse energy of ~1.1 mJ, a pulse length of 5 ns, and operating at 100 Hz repetition rate. Optical excitation at 650 nm was provided by a Opta OPO (Model 355 I, 410–700 nm) pumped by a Spectra-Physics QuantaRay LabSeries 150 Nd:YAG laser, with an incident pulse energy of ~2.2 mJ, a pulse length of 7 ns, and operating at 10 Hz. Excitation at each wavelength also included a depolarizer (DPP25-A, Thorlabs) to avoid polarization effects. The temperature was controlled using a Lakeshore 332 temperature controller and a laboratory-built helium flow cryostat. Transients were recorded as the static magnetic field was swept and continuous-wave microwave irradiation was applied (samples were measured with a microwave power of 0.5 mW). Orientation-dependent transient EPR studies were carried out with the use of a goniometer. The samples were transferred to a nitrogen glovebox where they were placed inside of 5 mm outer diameter quartz EPR tubes and fixed in place using Teflon tape. Fixing the on-substrate thin film allowed for orientation studies in trEPR. The sample fixed inside the EPR tube was then transferred to a pumping station (using a custom adapter which keeps the sample tube in the inert glovebox nitrogen environment) before being pumped to $4 \times 10^{-4}$ mbar and flame sealed. EPR simulation in Fig. 7 was done using EasySpin[92].

## Data availability

The data presented in this article are available from the corresponding authors on reasonable requests.

## Code availability

The codes used in generating the simulated spectra are available at https://zenodo.org/record/6985207#.YvZeNXZBwoZ (https://doi.org/10.5281/zenodo.6985207).

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

## Acknowledgements

N.M. acknowledges CSIR and IISc Bangalore for a Senior Research Fellowship. S.P. thanks support from the EPSRC project Strategic University Network to Revolutionize Indian Solar Energy-SUNRISE (EP/ P032591/1) and the Department of Science and Technology, New Delhi, for a Swarnajayanti Fellowship and SERB, IRHPA grant. J.N. thanks DST-SERB India for funding under the Ramanujan Fellowship scheme (RJN-187/ 2017) and the SciNet HPC Consortium, Compute Canada for the computational resources. N.A.P and R.B. acknowledge funding by the Deutsche Forschungsgemeinschaft (DFG, German Research Foundation) under Germany´s Excellence Strategy—EXC 2008—390540038— UniSysCat. A.K. and J.D. acknowledge support from the Department of Atomic Energy (DAE), Government of India, under Project no. 12-R&D-TFR-5.10-0100.

## Author contributions

S.P. conceived the project. N.M. synthesized the samples. N.M., W.K., and N.A.P. fabricated the thin films and performed basic optical characterizations. P.K., D.M., and J.N. performed theoretical calculations. W.K. performed the fs-TA, TGPL, and temperature-dependent ns-PL experiments. A.K. performed ns-TA experiments. N.A.P. performed tr-EPR experiments and analyzed the tr-EPR results together with R.B. W.K. J.D., and A.J.M. led the analyses of the data with contributions from all authors. W.K., N.A.P., N.M., S.P,and A.J.M. wrote the paper with input from all authors.

## Competing interests

The authors declare no competing interests.
