## [Peer Review File · Nature Communications]

REVIEWER COMMENTS

Reviewer #1 (Remarks to the Author):

This manuscript studied the singlet fission of the aggregates of DPP derivatives with steady state and time-resolved transient optical spectroscopic methods, as well as time-resolved EPR spectra. $1(T1T1)$ pair was detected, but the main channel of formation of triplet is proposed to be via ordinary SO-ISC. The results are interesting, the manuscript can be considered after consideration.

page 5, the authors state that ' $1(T1T1)$ must quantitatively decay to the ground state', I am afraid this is not true. Otherwise SF will be rarely observed, if the diffusion of $1(T1T1)$ to free T1 state is completely inhibited;

The way the transient absorption spectra is unusual. I suggest the authors follow the ordinary way to present the TA spectra, for instance, the ground state bleaching should be negative, and the excited state absorption band should be positive. Otherwise it is misleading. The triplet state absorption spectra of the DPP derivative should be determined with nanosecond transient absorption spectra. Global fitting and target analysis should be carried out for the transient absorption spectra;

How did the authors determine the ZFS D parameter of the triplet state of DPP derivative as positive? The D value of the triplet state of a chromophore and its derivative may have different sign. The authors proposed a specific ESP patten for the triplet state formed by SF, but the ESP pattern proposed is different from some of the previous reported, such as J. Phys. Chem. Lett. 2018, 9, 5855–586, Nature Commun., 2017, 8, 15171

Reviewer #2 (Remarks to the Author):

This paper presents a series of new thought-provoking careful observations for the TDPP chromophore and represents a real advance. I particularly like the use of two different types of solids generated from the same molecular material (it would be even better if the crystal structures of the polymorphs could be obtained). It seems to me, however, that the manuscript would benefit from a more thorough consideration of the ways in which the results could be interpreted.

P. 3, bottom: According to the literature [Z. E. X. Dance et al., *J. Am. Chem. Soc.* 130, 830 (2008); B. S. Basel et al., *Nature Commun.* 8, 15171 (2017)], the “robust” EPR identification of triplets as originating from singlet fission is not at all as robust as the authors claim. The sign pattern is different from the pattern that results from intersystem crossing induced by spin-orbit coupling, but it is the same as the pattern that results from intersystem crossing that occurs by the radical-pair mechanism that is favored by charge-separated states.

P. 4, bottom: I am not familiar with the term “Hamilton receptor”. It is probably sufficiently uncommon that it deserves more of a literature reference, and I wonder if it needs to be used at all. The capitalization of B in the name of the TDPP derivative used in this study, here and elsewhere in the text, should be removed.

P. 5, bottom: “slow yields” probably should be “low yields”.

P. 7, top: The characterization and proof of purity of the newly synthesized compound, provided in Supplementary Note 1, is incomplete and would not be considered acceptable in leading organic chemistry journals. Elemental analysis and an IR spectrum are missing, and no assignment is provided for the mass spectrum peaks at higher molecular masses than the expected molecular ion peak. The incomplete information provided does not prove that the compound was pure. E.g., it would be equally compatible with a mixture of the desired compound with colloidal silica particles.

P. 10: I think that long narratives should be in the text rather than in figure captions as in Fig. 3, but recognize that this is a matter of editorial policy and personal taste.

P. 12 The authors invoke sample heterogeneity and the presence of more than one sub-ensemble within their samples to explain the presence of two concurrent observed behaviors, and this is very reasonable. In the following text, however, such type of explanation for the observed complexity is not invoked or discussed again, and it is not explained clearly why it is excluded. For instance, in the discussion of one vs. two emissive states on page 15, could heterogeneity not be invoked again? In general, I find the experimental data to be of high quality, but it seems to me that their interpretation and discussion follow one particular path and do not consider or refute others that appear equally likely at first sight. For example, could charge-separated states not play a more significant role in the kinetic scheme than is currently proposed? The photophysical processes in this material appear to be exceedingly complex and the impressive number of rate constants invoked may be amenable to possible interpretations that are currently not mentioned. One of the puzzles that remains without an answer in the discussion is the near absence of spin-orbit induced

intersystem crossing in solution (<1%) and its implied greatly enhanced significance in the solid. What is responsible for this? Or is it really proven beyond doubt?

My recommendation is to either justify more profoundly the interpretation of the observations that is being proposed as the only one that is reasonable by excluding others, or to recognize explicitly that others may exist and discuss them as well.

Reviewer #1

This manuscript studied the singlet fission of the aggregates of DPP derivatives with steady state and time-resolved transient optical spectroscopic methods, as well as time-resolved EPR spectra. $1(T1T1)$ pair was detected, but the main channel of formation of triplet is proposed to be via ordinary SO-ISC. The results are interesting, the manuscript can be considered after consideration.

We do appreciate the reviewer for the positive evaluation of our work and hope that our point-by-point responses below would clear the reviewer's concerns.

page 5, the authors state that ' $1(T1T1)$ must quantitatively decay to the ground state', I am afraid this is not true. Otherwise SF will be rarely observed, if the diffusion of $1(T1T1)$ to free $T1$ state is completely inhibited;

We think there has been a misunderstanding on this point, and we apologize if our original wording was unclear. With this statement we are not making a generalization for all singlet fission materials, as the reviewer seems to be concerned. Rather, this statement occurs at the end of the introduction where we summarize the specific findings of our work on DPP aggregates. In thin films of many singlet fission materials, it is indeed known that triplet diffusion from $1(TT)$ can drive free triplet formation (Grieco et al., *Adv. Funct. Mater.*, 2017, 27, 1703929. / Zhu and Huang, *J. Phys. Chem. Lett.* 2018, 9, 6502., etc.). However, our results for the HR-TDPP-TEG J-aggregate reveal a case where the dissociation process could happen but is not very efficient, with the result that the long-lived T_1 exciton we observed must be generated from a classic intersystem crossing, as evidenced by the trEPR results. To make this nuance clearer, we have removed 'must' from this sentence in the revised manuscript.

The way the transient absorption spectra is unusual. I suggest the authors follow the ordinary way to present the TA spectra, for instance, the ground state bleaching should be negative, and the excited state absorption band should be positive. Otherwise it is misleading.

We respectfully disagree with the reviewer's opinion that the way of plotting transient absorption data with differential transmittance, $\Delta T/T$, is uncommon or misleading. Although it is true that many researchers adapt ΔA for plotting transient absorption results, $\Delta T/T$ has also been used in many highly cited papers in the singlet fission field over the last decade (Wilson et al. *J. Am. Chem. Soc.* 2011, 133, 11830 / Musser and Liebel et al. *Nat. Phys.* 2014, 11, 352 / Stern et al. *Nat. Chem.* 2017, 9, 1205, / Walker et al. *Nat. Chem.* 2013, 5, 1019 / Bakulin et al. *Nat. Chem.* 2016, 8, 16 / Monahan et al. *Nat. Chem.* 2017, 9, 341 / Fallon et al., *J. Am. Chem. Soc.* 2019, 141, 13867 / etc.). This practice is sufficiently widespread that we think there should be no concern about a misreading of the data, particularly as all of our plots are explicitly labeled with $\Delta T/T$. In our opinion, the choice of unit in which to present such results

is an issue of preference, not a scientific error, thus we would like to maintain our position to use $\Delta T/T$ in the manuscript. However, to help potential readers who are familiar with the unit of ΔA , we have added an auxiliary clarification in the manuscript as follows.

Page 9 in the revised manuscript) "...negative photoinduced absorption (PIA) bands above 2.4 eV and below 1.7 eV (Fig. 2a, note that we present all the data with differential transmittance, $\Delta T/T$)."

The triplet state absorption spectra of the DPP derivative should be determined with nanosecond transient absorption spectra.

We thank the reviewer's suggestion for presenting the triplet absorption spectra of the system. In the revised Supplementary Information, we have added nanosecond transient absorption data of the J-aggregate film and discussion to support the argument that the triplet-like photoinduced absorption features identified in the NIR by femtosecond transient absorption are not from spectral artifacts. Additionally, a representative triplet spectrum from the ns-TA is directly compared to our femtosecond results in main-text Fig.3c, and we refer to this data in the main text as follows:

Supplementary Fig. 12 in the revised Supplementary Information)

Supplementary Fig. 12 ns-TA results of film 1 after photoexcitation at 2.38 eV with 764 μ J/cm². a averaged and b normalized ns-TA spectra at indicated delay times. c Normalized $\Delta T/T$ kinetic averaged over the NIR PIA from 1.7 to 1.35 eV. The kinetic was fitted by a triexponential model (black solid line). The NIR PIA kinetic obtained from fs-TA is overlapped

for comparison. d Normalized population kinetics estimated from combined fs- and ns-TA results. We approximately delineate several temporal regions where specific species dominate, based on the time constants from multiexponential fits.

Additional discussion in the revised Supplementary Information)

Supplementary Note 3. ns-TA results for film 1

We performed ns-TA to get further insight into the long-lived kinetics of near-infrared (NIR) PIA in film 1 beyond the time scale accessible by fs-TA (Supplementary Fig. 12). The overall shapes of ns-TA spectra are identical to those of the spectra obtained through fs-TA (Fig. 3c): Two GSB peaks around 2.1 and 1.9 eV, overlapping PIA from 2.3 to 1.7 eV, and NIR PIA from 1.7 to 1.35 eV. Due to the possible contribution of thermal artifacts discussed in the main text, we only focused on the dynamics of the NIR PIA. The normalized $\Delta T/T$ kinetics of the NIR PIA is shown in Supplementary Fig. 12c. We were able to fit the kinetics by a triexponential model with the time constants of 1.2 ns, 21 ns, and 1.1 μ s. The first time constant, 1.2 ns, is analogous to the results obtained through fs-TA and ns-PL (Fig. 3d and Supplementary Fig. 17), enabling us to assign it as the lifetime of $^1(TT)$. The last, 1.1 μ s, supported by the tr-EPR results can be attributed to SO-ISC triplets. The origin of the intermediate time constant (21 ns) is ambiguous since there is no additional data available to explore the nature of this species, but we can suggest two possibilities: 1) $^1(T...T)$ or 2) free triplets, T_1+T_1 . In particular, the reason why we can assign all the species observed in ns-TA as triplet-related species is based on the similarity of the spectral shapes in the entire time window. Nevertheless, regardless of the origin of the species, what we can say from the overall TA results is that $^1(TT)$ dissociation in the DPP thin film is highly inefficient if it happens at all, and the SO-ISC triplet yield is also very low judging by the substantial decay of total signal within the first few ns (Supplementary Fig. 12d). Further light could be shed on the mechanism of these longer-time processes through magnetic-field-dependent experiments to manipulate the spin evolution of TT , as suggested in the main text, but those are beyond the scope of this work.

Global fitting and target analysis should be carried out for the transient absorption spectra;

We also appreciate the reviewer's suggestion on the global target analysis of the TA data. It is true that many researchers generally embrace a sequential model of the global target analysis and assign each spectrum a specific electronic state in the decay pathway. This approach works well for a system which follows a well-defined sequential decay, or exhibits two well-defined parallel pathways. However, the excited-state dynamics in films 1 and 2 are more complex with a distribution of states/subensembles that exhibit a range of decay rates into $^1(TT)$ with branching ratios that cannot be evaluated from any independent measurements. The same analysis algorithms can of course still be applied, but the resulting spectra become less meaningful to interpret. Instead, the main outcome of global fitting or target analysis

would be a set of time constants. We have nonetheless performed such analysis on our data to determine what additional insights can be gained. As shown below, global target analysis produces time constants in close agreement with those discussed in the main text based on our analysis of the artefact-free near-infrared photoinduced absorption kinetics. Our analysis of those kinetics, together with the isosbestic points highlighted in Fig. 3 of the main text, allows us to obtain direct insight into population transfer processes between pairs of electronic states without any presuppositions about the connectivity between states. We see this as an advantage over model-based algorithms. Because the global target analysis results provide no additional information that is not achieved from our current analysis, we provide this analysis for review only but prefer not to incorporate it into the manuscript or SI.

Fig. R1 Global target analysis of transient absorption data for films (a) 1 and (b) 2 with a sequential model ($A \rightarrow B \rightarrow C \rightarrow D \rightarrow GS$ for film 1 and $A \rightarrow B \rightarrow C \rightarrow GS$ for film 2, GS – ground state).

How did the authors determine the ZFS D parameter of the triplet state of DPP derivative as positive? The D value of the triplet state of a chromophore and its derivative may have different sign. The authors proposed a specific ESP pattern for the triplet state formed by SF, but the ESP pattern proposed is different from some of the previous reported, such as *J. Phys. Chem. Lett.* 2018, 9, 5855–586, *Nature Commun.*, 2017, 8, 15171

The sign of the ZFS D parameter is assumed to be positive based on *Salvadori, E. et al. J. Mater. Chem. A* 2017, 5, 24335–24343, where they observe a recombination triplet in a PC70BM/TDPP blend film sample. Here they assume a positive D value based on the recombination of a charge transfer (CT) state preferentially populating the $M_s = 0$ sublevel (the high field eigenstate) of the triplet, in this case the triplet of the TDPP. The sign of D can be different between chromophores and their derivatives however the singlet spin density

based on our DFT calculations for HR-TDPP-TEG (Fig. 1c) is still predominantly located over the TDPP core. We would expect the triplet spin density to also predominantly be on the core and hence we do not expect a change in the sign of D. Furthermore, the sign of D for an ISC triplet when explaining trEPR spectra is not significant beyond being able to assign the canonical peaks in the ISC born triplet EPR spectrum with the molecular eigenstates Tx, Ty, Tz. The sign of D does not change the interpretation of the data.

The ESP for the SF born triplet in HR-TDPP-TEG based on preferential population of the $M_s=0$ triplet sublevel is aee/aae (where a is absorption and e is emission) this is the same as for *J. Phys. Chem. Lett.* **2018**, 9, 5855 and *Nat. Commun.* **2017**, 8, 15171. However, the overall spectral shape in the simulated spectrum in main text Fig. 8b, is indeed different to the usually observed SF pattern for example shown in *J. Phys. Chem. Lett.* **2018**, 9, 5855 and *Nat. Commun.* **2017**, 8, 15171. Both these references are studying Pentacene/TIPS-Pentacene derivative based systems. In TIPS-Pentacene the ZFS E parameter is small and therefore the X and Y canonical peaks are close together. In the case of HR-TDPP-TEG or TDPP in general the E parameter is close to or approaching the 1/3 D value and therefore the highly rhombic triplet spin density causes the unusual spectral shape in the simulation for a TDPP based singlet fission spectrum. The difference in the E parameter results in the specific EPR spectrum for a SF born triplet as shown in Supplementary Note 6.

Reviewer #2

This paper presents a series of new thought-provoking careful observations for the TDPP chromophore and represents a real advance. I particularly like the use of two different types of solids generated from the same molecular material (it would be even better if the crystal structures of the polymorphs could be obtained). It seems to me, however, that the manuscript would benefit from a more thorough consideration of the ways in which the results could be interpreted.

We appreciate the reviewer for the positive evaluation of our work in terms of making two J- and H-type aggregates with the same material. We also hope that our point-by-point responses below would clear the reviewer's concerns.

P. 3, bottom: According to the literature [Z. E. X. Dance et al., J. Am. Chem. Soc. 130, 830 (2008); B. S. Basel et al., Nature Commun. 8, 15171 (2017)], the "robust" EPR identification of triplets as originating from singlet fission is not at all as robust as the authors claim. The sign pattern is different from the pattern that results from intersystem crossing induced by spin-orbit coupling, but it is the same as the pattern that results from intersystem crossing that occurs by the radical-pair mechanism that is favored by charge-separated states.

We agree with the reviewer that recombination of a charge separated state to a molecular triplet state in a donor-acceptor system leads to the same electron spin polarization pattern as for free triplets formed via singlet fission. Thus, the polarization pattern only allows a robust distinction between spin-orbit coupling induced ISC triplets from recombination and singlet fission triplets. If charge recombination can be ruled out, as here by the absence of any charge transfer state signal in the transient absorption and the only very weak charge transfer state signals in trEPR (at low temperatures), then it is nevertheless reasonable to assign the triplet trEPR spectrum consisting of an aee/aae polarization to singlet fission. However, the triplet state polarization observed in our case clearly deviates from the pattern expected for singlet fission but is only in agreement with formation by spin-orbit coupling induced ISC. Therefore, this line of argument is unnecessary, and we have reworded the relevant sentence as follows.

Page 3 bottom in the revised manuscript) "In particular, the spin polarization patterns of triplets born through SF carry signatures sharply distinct from those of spin-orbit coupling mediated intersystem crossing (SO-ISC) born triplets. Thereby facilitating a clear distinction between SF and SO-ISC born triplets."

P. 4, bottom: I am not familiar with the term "Hamilton receptor". It is probably sufficiently uncommon that it deserves more of a literature reference, and I wonder if it needs to be used at all. The capitalization of B in the name of the TDPP derivative used in this study, here and

elsewhere in the text, should be removed.

The name Hamilton receptor was named after Andrew D. Hamilton (*J. Am. Chem. Soc.* **1988**, *110*, 1318.). Further extensive studies on Hamilton receptor-based system were done by Jean Marie Lehn and others (*Chem. Eur. J.* **2002**, *8*, 1227.) as well. We have cited references of the Hamilton receptor in the main manuscript, wherever necessary.

[General] P. 5, bottom: “slow yields” probably should be “low yields”.

We thank the reviewer for pointing out the typo. We corrected it in the revised manuscript.

[Synthesis] P. 7, top: The characterization and proof of purity of the newly synthesized compound, provided in Supplementary Note 1, is incomplete and would not be considered acceptable in leading organic chemistry journals. Elemental analysis and an IR spectrum are missing, and no assignment is provided for the mass spectrum peaks at higher molecular masses than the expected molecular ion peak. The incomplete information provided does not prove that the compound was pure. E.g., it would be equally compatible with a mixture of the desired compound with colloidal silica particles.

We appreciate the reviewer’s concern, and apologize for the oversight in verifying the purity of our compound. To address the reviewer’s concern about the purity of the compound, we have done the analysis of isotope distribution the MALDI-TOF experiment (Fig. R2, Supplementary Fig. 4) and assigned all the high mass peaks. We found that the observed value (m/z for $[M]^+ = 1724.848$) is well-matched with the calculated one (1724.719). Also, the isotopic distribution pattern from MALDI-TOF experiment data exactly matches with the simulated data (Fig. a). This is updated in the revised Supplementary Information. In particular, the higher molecular mass peak is the Na and K ion adduct peak $[M+Na]^+$ and $[M+K]^+$, not from impurity. Such Na adduct peak in mass is reported in the literature (Lee et al. *J. Am. Soc. Mass Spectrom.* **2016**, *27*, 1491.). The isotopic distribution pattern of Na and K ion also exactly well-matches with our simulation data (shown in Fig. c & d). Additionally we have repeat the mass experiment where also we can see the $[M]^+$ and $[M+Na]^+$ peak (Fig. R2e). To further check the purity, we have performed concentration-dependent proton NMR experiments in d^8 -THF solvent. Because of Hamilton receptor groups, HR-TDPP-TEG tends to form aggregates (Cate et al. *J. Am. Chem. Soc.* **2004**, *126*, 3801.). We could find that upon dilution, the broad peaks in the aromatic region start shifting to the downfield and several peaks becomes well-resolved (shown in stack plot Fig. R3a). At 50 μ M concentration, no aggregation peaks are present (Fig. R3b) as well as there is no evidence of impurity peaks. All the peaks in the aromatic region of the spectrum (50 μ M) are assigned based on J-coupling and spin-spin splitting patterns, shown in Fig. R3b. In the FTIR spectrum (Fig. R3c), the broad peak ~ 3160 - 3640 cm^{-1} is associated with the amide -N-H, ~ 2848 - 2960 cm^{-1} -C-H, and 1676 cm^{-1} -C=O stretching frequency (Fig. R3d). With all the experimental evidence, we are confident to say that there is no issue with

the sample purity.

Fig. R2 (Supplementary Fig. 4a). MALDI-TOF mass spectrum of HR-TDPP-TEG.

Fig. R2b-d. (Supplementary Fig. 4b-d). Shows the isotope distributions of (b) $[M]^+$, (c) $[M+Na]^+$ and (d) $[M+K]^+$ peaks for HR-TDPP-TEG. Left panel shows the zoom image of MALDI-TOF experiment and right panel is for simulated isotope distribution of corresponding peak.

Fig.2e. MALDI-TOF mass spectrum of HR-TDPP-TEG repeat experiment.

Fig. R3. (a) Concentration dependent $^1\text{H-NMR}$ spectra for HR-TDPP-TEG in $d^8\text{-THF}$. The top spectrum recorded in CDCl_3 at RT. (b) Aromatic protons of HR-TDPP-TEG are assigned, based on J-coupling and splitting pattern (50 μM). At 9.67 and 9.02 ppm singlet peaks represents the two unsymmetrical inner and outer amide peaks of Hamilton receptor.

Fig. R3. (c) ^1H NMR spectra of HR-TDPP-TEG at $50\ \mu\text{M}$ concentration. (d) Solid state FT-IR spectrum for HR-TDPP-TEG recorded in KBr plate.

Fig. R4. Stack FT-IR spectra of HR-TDPP-TEG, TDPP-TEG and HR. The spectra are recorded in KBr plate in transmission mode at room temperature.

P. 10: I think that long narratives should be in the text rather than in figure captions as in Fig. 3, but recognize that this is a matter of editorial policy and personal taste.

According to the reviewer's suggestion, we reduced the length of captions in Fig. 3.

P. 12 The authors invoke sample heterogeneity and the presence of more than one sub-ensemble within their samples to explain the presence of two concurrent observed behaviors, and this is very reasonable. In the following text, however, such type of explanation for the observed complexity is not invoked or discussed again, and it is not explained clearly why it is excluded. For instance, in the discussion of one vs. two emissive states on page 15, could heterogeneity not be invoked again?

The reviewer raises an excellent point. Whereas we interpreted our transient absorption data with a parallel relaxation scheme involving simultaneous relaxation from different Frenkel exciton subensembles to $^1(\text{TT})$ at different rates, we described our transient PL spectroscopy in much simpler terms, as only FE vs $^1(\text{TT})$. This is a consequence of the very different timescales probed by the initial experiments. The ns-PL spectroscopy revealing $^1(\text{TT})$ emission can only provide a time-averaged spectral response before ~ 1 ns, meaning that any heterogeneity in the singlet emission is washed out. By the time delays over which we can distinguish unique emissive species in that instrument, our transient absorption kinetics have converged to a single, well-defined time constant, i.e. there is no further evidence of heterogeneity. To probe this behavior further, we have performed new fs-PL measurements with comparable resolution to the transient absorption data. In these new results, we observe similar time constants to those reported in TA and they are accompanied by slight spectral shifts and changes in the emissive spectral shape. That is to say, our PL data now fully supports the same model of a heterogeneous distribution of singlet states that exhibit different conversion rates into $^1(\text{TT})$. Those new results are incorporated into the main text as Fig. 4 with accompanying discussion. We note that the question of heterogeneity still cannot be straightforwardly addressed in the tr-EPR experiments. Because the population of long-lived states is so small, we cannot distinguish if they belong to different subensembles or whether the system has relaxed/excitations have migrated to similar geometries by the microsecond timescales we access there. Importantly, those measurements are completely blind to the heterogeneity in the initial singlet population that we can observe optically.

Page 12-13 in the revised manuscript) Transient grating PL (TGPL) results for film 1 provide further insight into the population kinetics of the lower Frenkel state (Fig. 4). To secure a sufficient signal-to-noise ratio of the data, we performed the experiments with a high pump fluence of about $50 \mu\text{J}/\text{cm}^2$ and confirmed that the integrated PL kinetics are in line with the averaged PIA kinetics in TA obtained with the same pump fluence (Fig. 4b), suggesting that we unambiguously track the kinetics of the same excited-state species, i.e. lower Frenkel. The higher pump fluence data ($240 \mu\text{J}/\text{cm}^2$) additionally support that the sub-ps dynamics stem

from singlet-singlet annihilation. After the singlet annihilation, biexponential decays with 8 and 47 ps were obtained through the multiexponential fitting, which is in good agreement with the time constants observed in the TA data. The slower component (300 ps) was not detectable due to the limit of the delay stage. During the 8- and 47-ps decays, we observe no intensity rise in the kinetics anywhere in our detection region. This allows us to rule out relaxation along the same exciton manifold, which typically presents as a dynamic redshift in time-resolved PL spectra with a decay at higher energy and a rise at lower energy.⁶¹ We note, however, that the peak positions (~ 10 meV difference) and vibronic peak ratios on these characteristic timescales clearly differ (Fig. 4c), implying a slightly different electronic structure between the two. Considering as well the substantial quenching in emission intensity that these decays reveal, and the close agreement with TA kinetics, we find that initial time constants reflect parallel 1(TT) formation from a distribution of lower Frenkel states.

Fig. 4 Lower Frenkel PL kinetics of film 1. **a.** TGPL map of film 1 following photoexcitation at 2.1 eV with a pump fluence of ~ 50 $\mu\text{J}/\text{cm}^2$. **b.** Comparison of TGPL (integrated PL) and TA (averaged PIA) kinetics with different pump fluences. The time constants in the panel were obtained by fitting the 50 $\mu\text{J}/\text{cm}^2$ integrated PL kinetics (yellow line) with a multiexponential function. **c.** Comparison of normalized PL spectra (light color – raw, dark color – smoothed) averaged over two different time regions (1-10 ps vs. 10-100 ps) in the data measured with 50 $\mu\text{J}/\text{cm}^2$. The inset shows the difference in peak positions between the two spectra.

In general, I find the experimental data to be of high quality, but it seems to me that their interpretation and discussion follow one particular path and do not consider or refute others that appear equally likely at first sight. For example, could charge-separated states not play a more significant role in the kinetic scheme than is currently proposed? The photophysical processes in this material appear to be exceedingly complex and the impressive number of rate constants invoked may be amenable to possible interpretations that are currently not mentioned.

We appreciate this concern from the reviewer, and we agree it is important to consider all reasonable interpretations of the data. We have addressed the point about heterogeneity in the PL data above. Regarding the large number of time constants, they do indeed present a potential concern and the photophysical pathway could be worryingly complex. However, we

are able to rely here on the spectral data from our PL and TA experiments. The TA data in particular reveals that we can describe all the phenomena in terms of just two types of state – singlets and triplets/triplet pairs. No spectral signatures of fundamentally different types of state were detected, whether in TA or PL. We can thus exclude the possibility of charge-separated states as directly populated, major participants in the photophysical pathway. We have further clarified in the main text our basis for the focus on only singlet and triplet-character states as follows.

Page 11 in the revised manuscript) We also explicitly highlight that we detect spectral evolution between only two electronic species, identified as either lower Frenkel or triplet-related states ($^1(TT)$ or T_1), over the entire experimental time window (Fig.3 and Supplementary Fig.12). This strongly implies that the direct population of other electronic species, e.g., charge-separated (radical anion and cation) states, in the excited-state dynamics in film 1 can be neglected.

One of the puzzles that remains without an answer in the discussion is the near absence of spin-orbit induced intersystem crossing in solution (<1%) and its implied greatly enhanced significance in the solid. What is responsible for this? Or is it really proven beyond doubt?

We realize here that we were insufficiently clear about the significance of ISC-formed triplets in the solid. We want to stress that the yield of long-lived triplets detected in tr-EPR is very small; it is a testament to the sensitivity of the technique that we are able to detect them. To reinforce this point, we have performed ns-TA of the J-aggregate film and tracked the triplet photoinduced absorption band out to comparable timescales. In the temporal regime accessible in tr-EPR, the triplet signal is a tiny fraction of what is detected on early timescales. Convolution the thermal artefacts makes precise yield determination difficult, but it appears likely that the triplet yield is comparably low ~1%. That said, it is useful as well to consider possible differences in ISC between solution and solid. It is well known that organic molecules in the monomeric state without any heavy metal atoms typically show very low ISC yield due to weak spin-orbit coupling strength, which is in line with the case of HR-TDPP-TEG monomer in the solution phase. But this is not always true, since several mechanisms, such as El-Sayed's rule, energy gap law, charge transfer mediation, and spin-vibronic coupling, can collectively or individually contribute to the overall ISC dynamics depending on molecular structures. Energetic resonance can play a very important role; for instance, ISC is relatively efficient in monomeric tetracene and scarcely detectable in TIPS-tetracene due to the stabilization of S_1 afforded by the TIPS groups, which breaks the S_1-T_1 resonance. In the solid phase (or stacked molecular dimers/oligomers), owing to π - π overlaps, intermolecular charge transfer interaction can exist. Because of this, strictly speaking, excited-state wavefunctions in aggregates are not purely excitonic, i.e., the contribution of charge-transfer states is inevitable

(Hestand and Spano, *J. Chem. Phys.* **2015**, *143*, 244707). Therefore, it is possible that the charge transfer state can accelerate ISC through the spin-orbit charge transfer ISC mechanism (Veldman et al. *J. Phys. Chem. A* **2008**, *112*, 5846. / Lefler et al. *J. Phys. Chem. A* **2013**, *117*, 10333.). And just as stabilization of the singlet sometimes breaks the resonance between singlet and triplet levels, we cannot rule out that aggregation may sometimes (as, potentially, in HR-TDPP-TEG J-aggregates) induce greater resonance between S_1 and a higher triplet level, thereby driving more efficient ISC. These are, however, just possible mechanisms, and with such a low triplet yield it is not possible to explore them in detail. Since those questions are beyond the scope of this work, we believe that the manuscript will not benefit from the inclusion of this speculative discussion. However, we have clarified the very low yield of long-lived triplets observed as follows.

Page 23-24 in the revised manuscript) Combining with ns-TA results, although it is not possible to precisely quantify the triplet yield due to the thermal artifacts, we highlight that the yield from the SO-ISC pathway is likewise evidently very small (Supplementary Fig. 12d).

My recommendation is to either justify more profoundly the interpretation of the observations that is being proposed as the only one that is reasonable by excluding others, or to recognize explicitly that others may exist and discuss them as well.

REVIEWERS' COMMENTS

Reviewer #1 (Remarks to the Author):

The revision is satisfactory, I suggest to accept the manuscript for publication in its current form.

Reviewer #2 (Remarks to the Author):

The authors have responded very satisfactorily to my questions and comments with a single exception: for reasons that I do not understand, they have not obtained the easiest standard check of purity: an elemental (combustion) analysis of their compound, which is simple, cheap, fast, and unambiguous. It is excellent that they added more spectral evidence, but none of it addresses a question I raised - how do they know that their material does not contain a small amount of colloidal silica or another impurity that yields no proton signals in NMR? A suspicious reader might wonder whether they did obtain an elemental analysis and it failed to agree with expectations. Except for this one odd issue, the manuscript is ready for acceptance. It is a very nice piece of work.

Reviewer #2 (Remarks to the Author):

The authors have responded very satisfactorily to my questions and comments with a single exception: for reasons that I do not understand, they have not obtained the easiest standard check of purity: an elemental (combustion) analysis of their compound, which is simple, cheap, fast, and unambiguous. It is excellent that they added more spectral evidence, but none of it addresses a question I raised - how do they know that their material does not contain a small amount of colloidal silica or another impurity that yields no proton signals in NMR? A suspicious reader might wonder whether they did obtain an elemental analysis and it failed to agree with expectations. Except for this one odd issue, the manuscript is ready for acceptance. It is a very nice piece of work.

We appreciate the reviewer's concern and apologize for the oversight in verifying the purity of our compound. At that moment our CHNS elemental analyzer was down, and we believed our higher molecular mass assignment (M, M+Na, M+K) with isotope distribution will be sufficient to comment on purity and clarify the reviewer's concern. However, we realise that small colloidal silica impurities might not show up in MALDI-TOF and we understand it is better to perform elemental (combustion) analysis as the reviewer suggested. To address the reviewer concern we have conducted the CHNS elemental analysis of dried powder sample, and we observe that the experimental result well-matches with theoretically predicted percentage within the experimental error bracket. Elemental analysis calculated (%) for $C_{92}H_{104}N_{14}O_{16}S_2$: C 64.02, H 6.07, N 11.36, S 3.71; found: C 63.86, H 6.08, N 11.25, S 3.75. Hence, we are confident to say that the compound is pure there is no colloidal silica present in the sample. This result we have added in the revised supplementary information.

Instrument:

Model: Thermo Scientific Flash 2000 Organic Elemental Analyzer
Analysis Mode: CHNS/O Mode
Software: Eager Experience

Experimental Conditions:

Detector : Thermal Conductivity Detector
Carrier gas : He
Reference gas : He
Column: CHNS/NCS column PQS SS 2M 6X5 mm in oven at 65° C (for CHNS)
Column: SS 1M 6X5mm (MS 5A) in oven at 65°C (for Oxygen)

Program:

CHNS furnace temp (°c) : 950
Oxygen furnace temp (°c) : 1060
Oven temp (°c) : 65
Oven run time (Sec) : 720 sec (for CHNS)
400 sec (for Oxygen)
Carrier gas flow (mL/min) : 140
Oxygen gas flow (mL/min) : 250
Reference gas flow (mL/min) : 100
Sampling delay (sec) : 12
Oxygen Injection end (sec) : 5
Auto zero : on
Detector gain : 1

Instrument Error : 0.22

Sample	Nitrogen %	Carbon %	Hydrogen %	Sulphur%	Oxygen%
HR-TEG	11.246198463	63.85923416	6.082651138	3.750001192	-